

# Spatial variation and linkages of soil and vegetation in the Siberian Arctic tundra – coupling field observations with remote sensing data

Juha Mikola[1], Tarmo Virtanen[2], Maiju Linkosalmi[3], Emmi Vähä[1], Johanna Nyman[4], Olga Postanogova[5], Aleksi Räsänen[2,6], D. Johan Kotze[7], Tuomas Laurila[3], Sari Juutinen[2], Vladimir Kondratyev[5], Mika Aurela[3]

[1]Ecosystems and Environment Research Programme, Faculty of Biological and Environmental Sciences, University of Helsinki, Niemenkatu 73, FI-15140 Lahti, Finland
[2]Ecosystems and Environment Research Programme, Faculty of Biological and Environmental Sciences and Helsinki Institute of Sustainability Science (HELSUS), P.O. Box 65, FI-00014 University of Helsinki, Finland
[3]Finnish Meteorological Institute, P.O. Box 503, FI-00101 Helsinki, Finland
[4]Ecosystems and Environment Research Programme, Faculty of Biological and Environmental Sciences, P.O. Box 65, FI-00014 University of Helsinki, Finland
[5]Yakutian Service for Hydrometeorology and Environmental Monitoring, Tiksi, Russia
[6]Department of Geography, Norwegian University of Science and Technology, NO-7491 Trondheim, Norway
[7]Ecosystems and Environment Research Programme, Faculty of Biological and Environmental Sciences and Helsinki Institute of Sustainability Science (HELSUS), University of Helsinki, Niemenkatu 73, FI-15140 Lahti, Finland

*Correspondence to*: Mika Aurela (mika.aurela@fmi.fi)

**Abstract.** Arctic tundra ecosystems will have a key role in future climate change due to intensifying permafrost thawing, plant growth and ecosystem carbon exchange, but monitoring these changes may be challenging due to the heterogeneity of Arctic landscapes. We examined spatial variation and linkages of soil and plant attributes in a site of Siberian Arctic tundra in Tiksi, northeast Russia, and evaluated possibilities to capture this variation by remote sensing for the benefit of carbon exchange measurements and landscape extrapolation. We distinguished nine land cover types (LCTs) – bare soil, lichen tundra, shrub tundra, flood meadow, graminoid tundra, bog, dry fen, wet den and water – to classify the variation in our site. To characterize the LCTs, we sampled 92 study plots for plant (biomass and leaf area index, LAI) and soil (organic matter OM%, bulk density, moisture, pH, litter layer depth, litter mass loss, temperature and active layer depth) attributes in 2014. Moreover, to test if variation in plant and soil attributes can be detected using remote sensing, we produced a normalized difference vegetation index (NDVI) and topographical parameters for each study plot using three very high spatial resolution multispectral satellite images (QuickBird and WorldView-2, portraying vegetation at 180, 220 and 750 growing degree days, DD with 0 °C threshold) and a digital elevation model (derived from a WV-2 stereo-pair image). We found that soils in our site ranged from mineral soils in bare soil and lichen tundra (on average 3.9 % OM) to soils of high OM% in graminoid tundra, bog, dry fen and wet fen (38 %), with shrub tundra and flood meadow being intermediate (21 %). Soil OM content was on average 14 g dm$^{-3}$ in bare soil and lichen tundra and 89 g dm$^{-3}$ in other LCTs. Total moss biomass varied from 0 to 820 g m$^{-2}$ among LCTs and high moss mass was associated with high soil OM%, except that wet fens with high OM% sustained low moss mass. Total vascular shoot mass was 7 g m$^{-2}$ in bare soil, on average 53 g m$^{-2}$ in lichen tundra and dry fen and 91 g m$^{-2}$ in other LCTs. Vascular LAI was on average 0.12 in bare soil and lichen tundra, 0.50 in bog, dry fen, shrub tundra and graminoid tundra, and 0.90 in flood meadow and wet fen. In late summer, soil temperatures at 15 cm depth were on average 14 °C in bare soil and lichen tundra, 9 °C in flood meadow and wet fen and 6 °C in other LCTs. Depth of the active soil layer doubled from early July to middle August, when it reached on average 42 cm in flood meadow and wet fen, 35 cm in bog, dry fen and graminoid tundra and 26 cm in bare soil and lichen and shrub tundra. When contrasted across study plots, total moss biomass was positively associated





with soil OM% and OM content and negatively with soil temperature, explaining 14–34 % of variation in soil attributes. Vascular shoot mass and LAI were also positively associated with soil OM content, and LAI with active layer depth, but the amount of variation explained was significantly lower (6–15 %). NDVI captured variation in peak season vascular LAI better than variation in moss biomass, but the difference depended on the phase of the growing season in the image:

180-DD, 220-DD and 750-DD NDVI captured 23, 17 and 7 % of moss mass variation and 25, 34 and 50% of vascular LAI variation, respectively. For this reason, soil attributes associated with moss mass were better captured by early season NDVI and those associated with LAI by late season NDVI. Topographic attributes were related to LAI and many soil attributes, but not to moss biomass and they could not increase the amount of spatial variation explained in plant and soil attributes above that achieved by NDVI. Our results illustrate a typical tundra ecosystem with great fine-scale spatial

variation in both plant and soil attributes. Mosses dominate plant biomass and control many soil attributes, including OM% and temperature, but variation in moss biomass is difficult to capture by remote sensing reflectance or topography. This suggests that using simple reflectance indices and DEM for spatial extrapolation of those vegetation and soil attributes that are relevant for regional ecosystem and global climate models warrants further inspection. Meanwhile, land cover maps of LCTs and their attributes, derived from remote sensing data and effective field sampling in different LCTs,

seem to provide the most reliable way to extrapolate vegetation and soil properties within Arctic tundra landscapes.

## 1 Introduction

Due to low temperatures that hinder the decomposition and mineralization of nutrients, the Arctic tundra is characterized by both high amounts of soil organic carbon (Hugelius et al., 2014) and very low primary production (Chapin, 1983). Climate warming is, however, increasing the rates of decomposition, nutrient mineralization and plant growth in northern

ecosystems (Hobbie, 1996; Tape et al., 2006; Schuur et al., 2009; Beermann et al., 2017; Commane et al., 2017) and monitoring these changes, and understanding the mechanisms that operate in the background, are necessary to assess the role of Arctic ecosystems in the future progress of climate change (Sitch et al., 2007; Myers-Smith et al., 2011). Remote sensing provides an effective means for field monitoring by linking surface features, such as vegetation characteristics, with local measurements of ecosystem carbon exchange (Marushchak et al., 2013, 2016; Sturtevant and Oechel, 2013).

Interpreting such data requires a good understanding of the spatial heterogeneity of vegetation and soil in the site of interest, but in many parts of the Arctic, such as the remote Siberian tundra, this knowledge is mostly lacking.

Arctic ecosystems can locally be very heterogeneous and comprise several intermingled plant communities (Virtanen and Ek, 2014; van der Wal and Stien, 2014). Soil properties can also vary considerably within landscapes (Suvanto et al., 2014; Siewert et al., 2016). To extrapolate local carbon exchange into wider areas, a study area is typically

categorized into land cover types (LCTs) using remote sensing methods supported by visual judgement of plant species composition and coverage (e.g. Marushchak et al., 2013). The classification criteria are rarely statistically judged, however, and the spatial variation of those plant and soil attributes that cause differences in carbon exchange among LCTs is seldom described in detail. In most cases, plant communities do not have sharp boundaries (e.g. Fletcher et al., 2010) and there is necessarily a lot of spatial variation within the LCTs as well. This variation is hardly ever described and a

key question is how well the obtained LCTs represent variation in the functional plant and soil attributes within heterogeneous landscapes such as the Arctic tundra.

Leaf area index (LAI) is commonly used to explain ecosystem carbon dynamics because it correlates well with the rate of plant photosynthesis (Aurela, 2005; Lindroth et al., 2008; Marushchak et al., 2013). LAI can also be mapped using remote sensing indices sensitive to green leaf pigments, such as the normalized difference vegetation index (Rouse



et al., 1973; Laidler and Treitz, 2003). However, soil attributes can be equally important in ecosystem carbon exchange (Euskirchen et al., 2017), and it is crucial to find out how soil attributes co-vary with those plant attributes that can be detected using remote sensing. Moreover, while LAI is a suitable measure of photosynthetically-active biomass for vascular plants, few studies have produced LAI estimates for mosses (Bond-Lamberty and Gower, 2007) and the

abundance of mosses has been estimated as areal coverage or thickness of the active green layer (Douma et al., 2007; Riutta et al., 2007). Capturing the spatial variation of moss biomass by satellite imagery can also be more difficult than capturing the variation of vascular plant biomass and LAI since mosses, in many cases, are covered by vascular plants (Bratsch et al., 2017; Liu et al., 2017). This can be a significant limitation in the monitoring of changes in Arctic carbon exchange since mosses are an important component of Arctic plant communities (Shaver and Chapin, 1991; Turetsky,

2003; Street et al., 2012). Mosses gather biomass in cold and moist areas, and besides participating in carbon assimilation (Moore et al., 2002; Turetsky, 2003; Street et al., 2012) they increase accumulation of carbon stocks in the soil as peat (Gorham, 1991) and control carbon release by isolating the permafrost soil from warm summer air (Beringer et al., 2001; Gornall et al., 2007).

The ability of satellite images to capture spatial variation in plant abundances and LAI depends on the phase of

growing season at the time of image acquisition (Langford et al., 2016; Juutinen et al., 2017). It is therefore likely that timing of imaging can also affect its ability to capture spatial variation of those soil attributes that are associated with vegetation attributes, but to our knowledge this has not been tested before. Observations of soil attributes that are linked to certain plant functional groups, such as mosses (better visible in the early season) and graminoids (abundant during mid- and late season only), could particularly rely on timely imagery. Observations of field attributes that are hard to

detect using reflectance indices might also benefit from being reinforced by site topography (Suvanto et al., 2014; Emmerton et al., 2016; Riihimäki et al., 2017). At any field site, vegetation and the soil interact continuously, and reciprocally affect the development of each other's attributes. At the landscape level, however, topography often guides the initiation and development of LCTs, with e.g. low elevation, wetter sites and high elevation, dryer sites having contrasting plant and soil dynamics. Using small-scale topography data might in such cases enhance the capture and

extrapolation of plant and soil variation across the landscape.

In this study, we examine the spatial variation and linkages of soil and vegetation characteristics in the Siberian Arctic tundra alongside opportunities of capturing the variation using multitemporal very high spatial resolution (VHSR) satellite imagery. Our specific targets are: (1) to produce a land cover map of our study area and describe the variation of vegetation and soil properties within and among the LCTs; (2) to test how well the ground-based visual judgement of

study plots into LCTs is supported by multivariate tests of their difference in functional attributes; (3) to test if the spatial variation in soil properties can be explained by the variation in plant abundance, and in particular the abundance of mosses vs. vascular plants; (4) to quantify the amount of variation in plant abundance and soil properties that can be captured by remote sensing indices and to test if images that portray the vegetation in different growth phases differ in their ability to capture this variation; and (5) to test if detailed topographic data could be used to enhance the capture of field variation

and improve the extrapolation of plant and soil attributes at the landscape level.

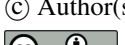



## 2 Materials and methods

### 2.1 Field site, land cover types and study plots

In 2010, a micrometeorological station was established ca. 500 m from the Arctic Ocean near the Tiksi Hydrometeorological Observatory, northeast Russia (71.59425° N, 128.88783° E) to provide eddy covariance (EC) measurements of the Arctic ecosystem-atmosphere exchange of $CO_2$, $CH_4$, $H_2O$ and heat (Uttal et al., 2016). The landscape at the site consists of lowlands and gently sloping hillslopes with highest elevations at 200–300 m. Around the EC mast, the landscape is relatively flat with some microtopographic variation, a 2–3° upward slope towards the north and a small stream, which runs through the area. During 1981–2010, the mean annual temperature in the area was -12.7 °C and the precipitation 323 mm (Arctic and Antarctic Research Institute AARI, 2016). Mean air temperatures for January and July were -30.2 and 7.7 °C, respectively, and the growing season had a mean heat sum of 668 DD (0 °C threshold) and lasted on average from 7 June to 26 September (AARI, 2016). At the EC mast, air temperatures follow temperatures measured at the climatological station closely. Bedrock in the site is composed of alkaline sandstone, mudstone and shale, and the soil is in continuous, deep permafrost (Grosswald et al., 1992). Seasonal fluctuations of temperature at the upper soil layer (0-40 cm, measured using individually calibrated PT100 sensors) follow fluctuations in air temperature (Fig. 1), but this pattern also markedly varies within the landscape. In bare soil areas, soil temperatures are relatively high in summer and low in winter, and at a depth of 5 cm the temperature is tightly coupled to air temperature (Fig. 1). In vegetated areas, the difference between summer and winter temperatures is less extreme and there is a clear lag between air and soil temperature (Fig. 1).

To provide information on vegetation and soil characteristics for the EC measurements (Aurela et al., *manuscript*; Tuovinen et al., *manuscript*), a field survey was carried out around the EC mast (covering ca. 1 km$^2$) in 2014. Nine land cover types (LCTs) – water, bare soil, lichen tundra, shrub tundra, flood meadow (along the stream), graminoid tundra (no obvious peat formation), bog (characterized by dwarf shrubs, hummocks and *Sphagnum* mosses), dry fen and wet fen (characterized by *Carex* and various mosses) – were first distinguished using ground-based visual judgement (Fig. 2, Table 1). Altogether 92 study plots were then established; the majority of plots (84 plots) was placed along 16 compass points at regular distances of 25, 50, 75, 100, 150 and 250 m from the EC mast, while the additional plots were placed along a few compass points at distances of 300, 350 and 400 m to balance the number of plots in different LCTs and to reach the longest distances of EC measurement coverage. In each study plot, four subplots (each 45 cm × 45 cm) were established at a radius of 2 m from the plot midpoint. Plant taxa were listed for each subplot (only dicotyledonous species could by identified at species level), which were then classified into one of the LCTs. Plot midpoints were georeferenced using a Global Positioning System (GPS) device (accuracy 1–3 m) and a measuring tape.

### 2.2 Collecting field data

One subplot per plot was destructively harvested during the peak season of plant biomass (from 23 to 27 July; heat sum 343–374 DD) to estimate plant aboveground biomass and to calculate the vascular plant leaf area index (LAI). For these measurements, plants were classified into seven functional types following the modification of Chapin et al. (1996) by Hugelius et al. (2011): (1) *Sphagnum* mosses, (2) other mosses, (3) dwarf shrubs (mainly evergreen, but also *Arctostaphylos alpina* and *Dryas octopetala*), (4) *Betula nana*, (5) *Salix* species, (6) herbs and (7) graminoids. To measure vascular plant biomass and LAI, all vascular shoot mass was removed from each harvested subplot using scissors. To estimate moss biomass, a 5 × 5 cm subsample of the moss layer was collected from a few selected subplots in different LCTs, and using moss areal cover, the biomass of *Sphagnum* and other mosses were estimated for each harvested subplot.



To obtain vascular plant LAI, the harvested leaves were scanned (Canon MP Navigator EX scanner; Canon Inc., Tokyo, Japan) and the green area of the scanned images determined using the GNU Image Manipulation Program 2 (GIMP 2) software. After LAI measurements, plant material was dried (24 h, 85 °C) for dry mass estimates.

Soil properties were investigated in the harvested subplots two weeks later (9 - 14 August). Litter layer depth
(consisting of dead vascular plant and moss material of discernible structure) was measured and a 10-cm deep soil sample, with known volume, was collected below the litter layer. In stony soils (found in bare soil and lichen and shrub tundra LCTs), the soil sample was collected using a spoon, while in other LCTs, a $3 \times 3 \times 10$ cm sample was cut from the soil using a knife. The soil samples were weighed, dried (48 h, 85 °C) and reweighed to calculate soil water concentration and bulk density, while organic matter concentration (OM%) was determined as loss on ignition. Soil OM content (g dm$^{-3}$)
was calculated by multiplying OM% by bulk density. Stones bigger than 1 cm$^3$ were excluded from measurements of soil bulk density and water and OM concentration, but their volume was taken into account when calculating soil OM content. The percentage of these stones of soil volume varied between 71 and 75 % among bare soil plots, 54 and 88 % among lichen tundra plots and 0 and 77 % among shrub tundra plots. Other LCTs had no large stones in the sampled layer. To measure pH, soil samples were collected from the subplots. The soil was homogenized, 30 ml of soil was mixed with
distilled water to obtain a total volume of 80 ml, the mixture was shaken and allowed to settle for 30 min and pH was measured using a Langen Hach HQ 40d portable field device. Finally, to estimate relative differences in decomposition rate, two teabags (Lipton® Pyramid green tea; Keuskamp et al., 2013) were placed in one subplot per plot (from 1 to 5 July, 159–174 DD; two plots on 11 July, 236 DD) – one on the soil surface and another buried in the soil at a depth of 5 cm. Teabags were collected 31–41 days later (from 9 to 14 August, 519–583 DD) and dried (24 h, 85 °C) to estimate
mass loss (expressed as mass loss per day to control for the varying lengths of decomposition in different plots).

Depth of the biologically active (unfrozen) soil layer was estimated weekly in a non-harvested subplot in each plot from early July to mid-August (calendar weeks 27–33; 160–550 DD) using a sharpened iron rod. Soil temperature was simultaneously measured at a depth of 15 cm using an Amprobe TMD-50 Thermocouple K-type thermometer (Amprobe Instrument Corporation, Everett, USA), but due to malfunctioning of the meter during the latter part of data
collection, measurements are only available for weeks 27–31 (160–380 DD).

To facilitate later usage of field data, mean values of measured plant and soil attributes at the different LCTs are presented in Supplementary Table 1 in addition to being visualized in graphs.

### 2.3 Satellite images and remote sensing indices

To test how the spatial variation of plant and soil characteristics in our study area can be detected using satellite imagery,
we produced a reflectance index NDVI using three VHSR multispectral satellite images – i.e. one QuickBird image (QB, DigitalGlobe, Westminster, CO, USA; 15 July 2005) and two WorldView-2 images (WV-2, DigitalGlobe, Westminster, CO, USA; 12 August 2012 and 11 July 2015). WV-2 images had a resolution of 2 m and QB image was delivered as a pan-sharpened product with 0.6 m resolution. The selected images were free of clouds and portrayed the vegetation at different growth stages: the 2005 QB image during the early growing phase (180 DD; 0 °C threshold), the 2015 WV-2
image somewhat later (220 DD) and the 2012 WV-2 image during the late growing phase (750 DD). To enable comparison of images taken under different atmospheric conditions, the images were corrected for atmospheric scattering when necessary and transformed into surface reflectance values. The 2015 WV-2 image was delivered as an atmospherically compensated product (Digital Globe AComp), but for the QB and 2012 WV-2 images, we used the dark-object subtraction method (Chavez, 1988).



Reflectance values were extracted for each study plot (using a circular area of a 5 m radius) to calculate NDVI = (NIR - VIS) / (NIR + VIS), where NIR and VIS are the near-infrared and visible red regions of the spectral reflectance (Rouse et al., 1973). As green plant tissue absorbs VIS and reflects NIR, NDVI indicates the biomass and photosynthetic capacity of plant leaves, which can then be utilized in the remote sensing and spatial examination of LAI (Tucker, 1979; Laidler and Treitz, 2003). The NDVIs that were calculated using the 2005 QB and the 2015 and 2012 WV-2 image reflectance values are subsequently referred to as 180-DD, 220-DD and 750-DD NDVI, respectively.

**2.4 Digital elevation model**

To calculate topographical parameters, we constructed a 2 m resolution digital elevation model (DEM) using the panchromatic bands (50 cm resolution) of the 2015 WV-2 stereo-pair image. When building the point-cloud, we co-registered the images using 25 ground control points in precise locations such as in buildings and lake shorelines, 25 auto-tie points and the rational polynomial coefficient information of the images. We included the elevation information for some of the GCPs by visually interpreting the topographic map and the ASTER DEM of the area. Using the point-cloud, we then calculated the 2 m resolution DEM using linear interpolation and Delaunay triangulation. To remove artefacts from the DEM, we masked and filled the areas of water, used a slope-based filter (Vosselman, 2000) in SAGA-GIS 2.1.2 (Conrad et al., 2015) and manually removed some obvious artefacts. The DEM was constructed using Erdas Imagine 2014 (Intergraph, Huntsville, AL, USA) and post-processing was carried out in Erdas Imagine, ArcGIS 10.3.1 (Esri, Redlands, CA, USA) and SAGA-GIS 2.1.2 (Conrad et al., 2015).

From the DEM data, we calculated elevation, slope (in degrees), solar radiation (SR), topographic position index (TPI-25 and TPI-100, using 25 m and 100 m neighbourhood radii, respectively) and topographic wetness index (TWI) to test whether these attributes could help in catching spatial variation in vegetation and the soil through remote sensing. SR represents potential June-August solar radiation into each pixel using 30 min intervals, TPI is a measure of the relative altitudinal position of each pixel (Guisan et al., 1999) and TWI models potential soil moisture with the help of the upslope contributing area and the local slope. For TWI, we used a modification called SAGA wetness index, where high TWI values in flat areas are spread into larger neighbourhoods (Böhner and Selige, 2006). TPI and TWI were calculated using SAGA-GIS 2.1.2 (Conrad et al., 2015) and SR using ArcGIS 10.3.1 (Esri, Redlands, CA, USA). At our site, SR had a statistically significant correlation ($P \leq 0.05$) with elevation ($r = 0.30$, $P = 0.003$, $n = 92$), TPI-25 with TPI-100 ($r = 0.66$, $P < 0.001$), and TWI with elevation ($r = -0.34$, $P = 0.001$) and TPI-100 ($r = -0.30$, $P = 0.004$).

**2.5 Land cover classification and landscape estimates of plant and soil attributes**

Land cover was categorized into nine LCTs (seven plant community types along with bare soil and water) in an object-based setting using full-lambda schedule (FLS) segmentation and random forest (RF) classification in 2 m resolution. As plant communities differ in phenology (Juutinen et al., 2017), both WV-2 images were employed. The images were ortho-corrected with the help of the constructed DEM and co-registered using field measured GPS data, and in addition to the optical data, DEM-derived features were used for classification.

The co-registered images were first segmented using FLS in ERDAS Imagine 2014 (Intergraph, Madison, AL, USA). FLS segmentation is region-based and the pixels are merged with the help of spectral (mean pixel value in the segment), textural (standard deviation of pixel values in the segment), shape (areal complexity of the segment) and size information, which we weighted 0.7, 0.7, 0.3 and 0.3, respectively. The average size of the segment (i.e. pixel/segment ratio) was set to 50 (i.e. 200 m²). For each segment, 262 features were calculated using the image and DEM data. For





each image band, the mean and SD were calculated for each segment together with 13 grey-level co-occurrence matrix (GLCM) features (Haralick et al., 1973), which are among the most widely used textural features (Blaschke et al., 2014). When calculating the GLCM features, the data were quantized to 32 levels. In addition, means and SDs were calculated for three spectral indices – NDVI (Rouse et al., 1973), red-green index (Coops et al., 2006) and normalized difference

water index (McFeeters, 1996) – as well as for TWI, TPI-25, TPI-100, elevation and slope layers derived from DEM.

For the classification, we first built a training dataset using eight 150 m long transects with known transitions, collected in 2014. One LCT was set for one segment and to complement the transect data, we visually interpreted LCT for some segments that were easily interpretable. Overall, we had 19–50 training segments for each class. We then reduced the number of features from 262 to 109 using the RF-based wrapper feature selection algorithm Boruta (Kursa and

Rudnicki, 2010; Räsänen et al., 2014; Li et al., 2016) in R 3.2.2 (R Core Team, 2015). After 1000 RF runs in Boruta, variables were deemed confirmed, rejected or tentative, and if tentative, a tentative rough fix (Kursa and Rudnicki, 2010) was carried out. The segments were then classified using RF in the package randomForest (Liaw and Wiener, 2002) in R 3.2.2 (R Core Team, 2015). RF is an ensemble classifier, which combines multiple classification trees (Breiman 2001), handles multi-dimensional data well and is often valued as one of the best classifiers (Belgiu and Dragut, 2016; Rodriguez-

Galiano et al., 2012). As RF is relatively insensitive to parametrization (Rodriguez-Galiano et al., 2012), we used default parameter values. In each tree of RF classification, two-thirds of the data are used for training and one-third, the so-called out-of-bag (OOB) data, for testing. Because of the OOB data, cross-validation or external validation data are not necessary for RF classification. Nevertheless, in order to check if our classification also worked in the overall landscape, we calculated both internal (with the help of the OOB data) and external classification accuracy. For the pixel-based external

classification accuracy, we used the 92 field plots and 139 random points, calculated a 5 m radius for each point and cross-tabulated the field observation with the classification.

Using sample data means of different LCTs and taking into account the LCT distribution in the landscape, we finally calculated the landscape distribution and grand total of the vegetation and soil parameters. When estimating the OM content of the active soil layer for different LCTs and the landscape, we used OM content values of soil samples

collected 0-10 cm below the litter layer for the whole active layer.

## 2.6 Statistical analysis of the field data

To avoid a multitude of pair-wise comparisons and to provide easy statistical inference in the graphs (Cumming, 2009; Paaso et al., 2017), the statistical significance of differences in plant and soil attributes among LCTs and the seasonal trends in soil temperature and active layer depth were interpreted using 85% confidence intervals (85% CI) of LCT means.

In this approach, non-crossing CIs of two means denote a statistically significant difference between the means. Using 95% CIs is a more common approach, but too conservative for testing mean differences, and the best approximation of α = 0.05 is achieved using 85% CIs (Payton et al., 2000).

Non-metric multidimensional scaling (NMDS) ordination was used to represent whether the visually-judged LCTs differed in vascular plant species composition (only dicotyledonous species were included in the analysis) and plant

and soil functional variables (including all soil attributes, except for temperature and active layer depth, and biomass of the seven plant functional groups). The 92 study plots were used as sampling units. We used the Raup-Crick (plant species composition; presence/absence data) and Bray-Curtis (soil attribute and biomass data) coefficients as dissimilarity measures (vegan package in R, see Oksanen et al., 2017). To test whether the eight LCTs (excluding water) were significantly different in species composition of dicotyledonous plant species and plant and soil functional variables, a



multi-response permutation procedure (MRPP) was used. MRPP is a non-parametric permutation procedure for testing the hypothesis of no difference between groups (McCune et al., 2002), here the eight LCTs. MRPP returns a test statistic *T* that describes the separation between groups (the more negative *T*, the stronger the separation). The *mrpp* function in the vegan package (in R) was used to perform this test.

The ability of variation in moss biomass, vascular plant biomass and vascular plant LAI to explain the variation in soil attributes among the study plots (n = 92) was tested using linear regression. Since soil temperature and active layer depth were not measured at the same subplots as other plant and soil attributes, and since LCT differed between the subplots in 19 of the 92 plots (e.g., one subplot representing bog and the other dry fen), associations between plant attributes and soil temperature and active layer depth were tested using only those plots (n = 73) where the subplots

represented the same LCT. The ability of NDVI to capture variation in soil attributes was also tested using linear regression, and since the area used for extracting reflectance values covered both subplots of a study plot, all 92 plots were included in the analysis of soil temperature and active layer depth. The associations of the three NDVIs (calculated for the different phases of growing season using the three satellite images) with moss biomass and vascular plant LAI (measured at the moment of peak plant biomass) were analysed using logarithmic regression.

The association of topographic attributes of study plots with their plant and soil attributes were tested using Pearson's correlation analysis. The ability of topography to enhance the explanation of variation of plant and soil attributes among the study plots was then tested by comparing the coefficients of determination ($R^2$) of multiple regression models that included (a) those topographical attributes that significantly correlated (p < 0.05) with the dependent variable, (b) the best or worst NDVI, and (c) the best or worst NDVI amended by the topographic features used in (a).

## 3. Results

### 3.1 Variation of soil attributes among land cover types

The soils in our study area ranged from mineral soils in bare soil and lichen tundra LCTs to soils characterised by high OM% in graminoid tundra, bog and fen LCTs, with shrub tundra and flood meadow LCTs featuring intermediate values (Fig. 3a). Soil bulk density had an opposite pattern with the same grouping of LCTs (Fig. 3b). Soil OM content was

distinctly low in bare soil and lichen tundra, but differences among other LCTs were inconsequential and only shrub tundra significantly differed from graminoid tundra, bog and dry fen (Fig. 3c). The pattern in litter layer depth (Fig. 3d) loosely followed the pattern in soil OM%. Tea mass loss at the soil surface was higher in wet fen than other LCTs (Fig. 3e), whereas for buried tea, mass loss was higher in bare soil than in flood meadow and the three tundra types (Fig. 3f). Soil pH was highest in bare soil and lichen tundra, lowest in bog and intermediate in other LCTs (Fig. 3g). The pattern

and grouping of LCTs in soil water concentration (Fig. 3h) were a mirror image of those in bulk density.

Soil temperatures increased on average by 5 °C from early July (calendar week 27) to early August (week 31) (Fig. 4a). Temperature in bog soil increased steadily throughout the summer, but other LCTs had significant fluctuations and a transient low in early August (Fig. 4a). Throughout the summer, soil temperatures were highest in bare soil and lichen tundra, and although the other LCTs partly showed a mixed order, flood meadow and wet fen had higher soil

temperatures than other LCTs in most measurements (Fig. 4a). Depth of the active soil layer doubled and increased on average by 16 cm from early July to middle August (week 33) (Fig. 4b). Deepening was relatively stable through the summer except in bare soil and lichen tundra, which had no significant progress after early July, and in shrub tundra, where deepening stagnated in August (Fig. 4b). Unlike soil temperatures, the order of LCTs in the active layer depth was



reorganized during the summer: while bare soil, lichen tundra, flood meadow and wet fen all had a deeper active layer than other LCTs in early July, only flood meadow and wet fen had high values in mid-August, with bog, dry fen and graminoid tundra showing intermediate and bare soil, lichen and shrub tundra low values (Fig. 4b).

### 3.2 Variation of plant biomass and LAI among land cover types

Total biomass of mosses varied from 0 to 820 g m$^{-2}$ among the LCTs (Fig. 5a) and followed the LCT grouping in soil OM% (Fig. 3a), except that wet fens with high OM% sustained low moss biomass. *Sphagnum* mosses were abundant in graminoid tundra, bog and dry fen and mostly absent in other LCTs (Fig. 5b), whereas other moss species had no or low biomass in bare soil, lichen tundra and wet fen and high biomass in other LCTs (Fig. 5c). Total vascular shoot mass was low in bare soil, intermediate in lichen tundra and dry fen and equally high in the other LCTs (Fig. 5d). Total vascular

shoot mass exceeded or equalled total moss mass in bare soil, lichen tundra and wet fen, but was 60-90 % lower in the other LCTs (Fig. 5a, d). *Betula nana* and dwarf shrubs were mainly found in shrub tundra and bog (Fig. 5e, f), *Salix* in all other LCTs except bare soil, lichen tundra and wet fen (Fig. 5g) and herbs in the low OM% soils of flood meadow and lichen and shrub tundra (Fig. 5h). Graminoids dominated vascular shoot mass in flood meadow and wet fen, had equal biomass with other vascular plants in graminoid tundra and dry fen and were marginal in other LCTs (Fig. 5i). Leaf area

index (LAI) was low in bare soil and lichen tundra, intermediate in bog, dry fen and shrub and graminoid tundra, and high in flood meadow and wet fen (Fig. 6).

### 3.3 Plant community and functional differences among land cover types

The multi-response permutation procedure (MRPP) showed that LCTs differed significantly in both species composition of dicotyledonous plants (T = -14.818, P < 0.001) and plant and soil functional characteristics (T = -15.024, P < 0.001)

(Fig. 7). In both datasets, the gradient from low to high soil OM and water concentration emerged in the grouping of LCTs (Fig. 7) and most of the pair-wise comparisons of LCTs showed highly statistically significant differences (Table 2). However, bare soil did not differ from lichen tundra and wet fen did not differ from graminoid tundra and dry fen in species composition (Fig. 7a, Table 2). Likewise, graminoid tundra did not differ from flood meadow, bog and dry fen, and bog did not differ from dry fen in the analysis of functional attributes (Fig. 7b, Table 2). Spatial variation within LCTs

(i.e. among study plots, illustrated by the within-type delta) was substantially higher for functional characteristics than for species composition (Table 2).

### 3.4 Land cover classification and plant and soil OM masses at the landscape scale

In the land cover classification, internal and external classification accuracies were 80 and 49 %, respectively, and the user and producer accuracies varied between 10 and 100 % (Supplementary Table 2). Based on the LCT map (Fig. 8),

shrub tundra covers over one fourth of the landscape area, wet fen and bare soil ca. 15 % each, dry fen, lichen tundra and bog ca. 10 % each, and graminoid tundra and flood meadow together less than 5 % (Table 3). In comparison to these proportions, shrub tundra and particularly wet fen were responsible for a greater share of vascular leaf production, both producing one-third of landscape leaf area, while dry fen and bog followed their areal proportions (Table 3). This pattern among the four LCTs was also evident in vascular shoot mass and SOM, except that the share of shrub tundra was double

that of wet fen in biomass and the proportions of dry fen and bog were elevated in SOM (Table 3). Moss biomass deviated from this pattern, however, as shrub tundra, dry fen and bog each were responsible for ca. 30 % of landscape moss mass (Table 3). The amount of biologically active SOM doubled during the growing season, but the landscape distribution





remained mostly the same (Table 3). In comparison to the combined peak season vascular shoot and moss biomass, the quantity of biologically active SOM was 30- and 60-fold in early and late season, respectively (Table 3).

**3.5 Linkages between plant biomass, LAI and soil characteristics**

When tested across all field plots, variation in total moss biomass explained a significant proportion (14–34 %) of variation in litter layer depth, soil OM% and OM content, water concentration and late summer temperature (Fig. 9). Soil OM% and OM content, litter layer depth and water concentration were positively and temperature negatively associated with moss biomass (Fig. 9). Variation in moss biomass did not explain variation in tea mass loss rates or late summer active layer depth (Fig. 9). Vascular shoot mass was positively associated with SOM content ($R^2 = 0.12$, P = 0.001), but not with other soil characteristics (Supplementary Fig. 1), whereas vascular LAI was positively associated with SOM content, water concentration and late summer active layer depth (Fig. 10). Coefficients of determination for LAI (6–15 %) were, however, low in comparison to those for moss biomass. Across the field plots, moss biomass did not correlate with vascular shoot mass (r = -0.05, P = 0.674, n = 92), but had a weak negative correlation with vascular LAI (r = -0.20, P = 0.051).

**3.6 Capturing plant and soil variation using remote sensing indices and topography**

The three NDVIs captured variation in vascular LAI ($R^2 = 0.25$–0.50), measured at the peak season in the field, better than variation in moss biomass ($R^2 = 0.07$–0.23) (Fig. 11). However, this difference depended strongly on the phase of the growing season in the satellite image: i.e., the amount of variation of moss biomass captured by NDVI decreased and the amount of variation of LAI increased with the DD of the image (Fig. 11). Both early- and late-season NDVI were positively associated, through the vegetation signal, with SOM content, soil moisture, litter layer depth and active layer depth and negatively with soil temperature (Fig. 12). However, seasonal trends again emerged: the amount of variation of SOM, moisture and active layer depth captured by NDVI increased and the amount of variation of litter layer depth and soil temperature decreased with the DD of the image (Fig. 12).

Most of the correlations between the topographical features and plant and soil attributes were low and statistically non-significant. However, elevation correlated negatively with vascular plant LAI (r = -0.33, P = 0.001), soil OM content (r = -0.29, P = 0.005), soil moisture (r = -0.44, P < 0.001), soil active layer depth (r = -0.27, P = 0.009) and litter layer depth (r = -0.21, P = 0.05). Slope correlated positively with soil active layer depth (r = 0.21, P = 0.043) and temperature (r = 0.22, P = 0.033) and the wetness index positively with vascular plant LAI (r = 0.40, P < 0.001). Solar radiation was negatively linked to litter layer depth (r = -0.21, P = 0.041) and soil active layer depth (r = -0.24, P = 0.024), but not to plant attributes. Of the indices that describe relative elevation in the landscape, TPI-100 correlated negatively with vascular LAI (r = -0.21, P = 0.05) and TPI-25 positively with soil temperature (r = 0.21, P = 0.05). Moss biomass was not significantly related to any topographic attribute. The ability of topography alone to explain variation in vascular LAI and soil attributes was low in comparison to the best available NDVI predictor and amending the best NDVI predictor with topographic features only marginally improved the amount of explained variation, except for the active layer depth (Table 4). However, greater improvement was achieved, except for litter layer depth, when the worst NDVI predictor was supplemented with topographic features (Table 4).





## 4. Discussion

Our aim was to describe the spatial variation and linkages of soil and plant attributes at a Siberian Arctic tundra field site and to evaluate the possibility to capture this variation by remote sensing for the benefit of EC measurements of greenhouse gas fluxes and landscape extrapolation. We found high spatial variation at our site: the soils ranged from
mineral to organic and aboveground plant biomass varied greatly among the established land cover types (LCTs). This led to distinct seasonal dynamics of soil temperature and active layer depth among LCTs. On the other hand, our multivariate analysis suggests that not all LCTs differed significantly in those attributes that control ecosystem functioning. We also found that variation in soil attributes within the landscape was more closely linked to variation of moss biomass than to variation of vascular plant LAI, whereas remote sensing reflectance indices could far better capture
variation in vascular LAI. Moreover, because variation in moss biomass was better captured by early-season reflectance, timing of the image affected the capture of soil variation. For instance, variation in soil temperature, controlled by moss biomass, was better captured by early- than late-season image. Contrary to our expectations, site topography was not linked to variation in moss biomass and could not significantly enhance the capture of spatial variation of plant and soil properties above the level achieved by reflectance indices. Altogether, our field site exemplifies a typical tundra
ecosystem with great fine-scale spatial variation in plant and soil attributes. Mosses dominate plant biomass and control many soil attributes, but variation in moss biomass is difficult to capture by remote sensing reflectance. This suggests that a land cover map, or multiple maps of land cover, vegetation and soil attributes, derived from remote sensing data and accompanied by effective field sampling in different LCTs, are needed to extrapolate vegetation and soil properties within the landscape.

### 4.1 Field variation and linkages between vegetation and the soil

Soil temperatures are critical in the functioning of Arctic ecosystems. Permafrost and low temperatures are the main reason for slow nutrient mineralization (Callaghan et al., 2004; Ernakovich et al., 2014), which in turn limits primary production (Chapin, 1983), and ecosystem carbon exchange is strongly influenced by soil thawing and warming (Schuur et al., 2009; Commane et al., 2017). In our site, bare soil and lichen tundra had distinctly warmer soils than other LCTs
throughout the summer. This is most likely because bare soil and lichen tundra have a low albedo and lack larger plants that would reduce radiation input. Water has high specific heat efficiency and the low soil water content in bare soil and lichen tundra could also contribute to rapid warming, but this does not seem to be the case, since water content does not explain differences in soil warming in other LCTs. Instead, these differences seem to be explained by plant community structure. Those three LCTs – graminoid tundra, bog and dry fen – which produce high moss biomass, have a steady,
slow increase in soil temperature through the growing season. In contrast, those LCTs with lower moss mass – shrub tundra, flood meadow and wet fen – all display, despite very different soil water content, fluctuations in soil temperature that follow the form of those in bare soil and lichen tundra. These findings support the view that one of the main mechanisms through which mosses affect the functioning of Arctic ecosystems is isolating the permafrost soil from warm summer air (Beringer et al., 2001; Gornall et al., 2007) and suggest that climate warming in our site will least affect soil
functioning in the three LCTs of highest soil OM content. This conclusion is further supported by a thin biologically active soil layer in these LCTs, and our results fully support the idea that mosses both generate (Gorham, 1991) and conserve (Beringer et al., 2001; Gornall et al., 2007) carbon in Arctic soils. Aboveground plant biomass and particularly the spatial variation of biomass was also dominated by mosses: moss biomass varied 14.4-fold among the LCTs, while the vascular LAI varied 4.7-fold (these comparisons exclude bare soil, which had no mosses and very low LAI). While





differences in soil properties indicate great variation in carbon release, these differences in plant photosynthetic biomass indicate equally great variation in carbon assimilation among the LCTs.

Plant communities seldom have sharp boundaries in the field (e.g. Fletcher et al., 2010) and as expected, we found considerable within-LCT variation in plant species composition, soil attributes and plant biomasses. Despite this
variation, multivariate analyses suggested that differences among most LCTs were statistically highly significant, suggesting that those LCTs that were visually judged in the field were real. Some LCTs that were dominated by graminoids, i.e. graminoid tundra, dry fen and wet fen, could not be distinguished based on plant species composition, but this result is likely an artefact because we only used dicotyledonous species for analysing species composition. Similarities among LCTs in functional attributes (plant functional type biomass and soil attributes excluding temperature
and active layer depth) are instead real. Our analysis suggests that graminoid tundra does not differ from flood meadow, bog and dry fen in functional attributes, and neither does bog and dry fen differ from each other. Notably, this idea is supported by our finding that graminoid tundra, bog and dry fen had very similar soil temperatures and active layer depth. Altogether these results predict that some LCTs that can be distinguished by plant species composition and are even dominated by different plant functional types, like bog and dry fen, may not differ in functioning.

When plot-to-plot variation in soil attributes and plant production were contrasted, variation in SOM content was positively related to variation in all measures of plant production. These three measures – moss biomass, LAI and vascular plant biomass – differ greatly in the quality of litter they create. Moss litter is generally highly recalcitrant to decomposition (Coulson and Butterfield, 1978; Hobbie, 1996) and e.g. *Sphagnum* litter decomposers slower than *Carex* litter (Palozzi and Lindo, 2017). Leaf litter (a derivative of LAI) in turn decomposers faster than woody litter (Hobbie,
1996), which is a major component of vascular plant biomass in LCTs dominated by shrubs (Hobbie, 1996; Weintraub and Schimel, 2005). That all measures of plant production, regardless of their wide variation in litter quality, are equally positively related to SOM content suggests that low soil temperature rather than low quality of litter promotes the accumulation of SOM in our site. Earlier findings of vascular plants enhancing SOM accumulation in *Sphagnum* dominated peatlands (Andersen et al., 2013) further supports this idea; i.e. in adverse conditions, litter of higher quality
can also contribute to SOM accumulation.

In contrast to variation in SOM content, variation in other soil attributes was mostly related to variation in moss biomass only. Soil OM%, litter layer depth, moisture and temperature were clearly connected to moss biomass. The positive association of moss biomass with soil water content partly tells of the habitat requirements of mosses, but the positive association with soil OM% and litter layer depth and the negative association with soil temperature once again
demonstrate the ability of mosses to both generate SOM (Gorham, 1991) and insulate it from warm air temperatures (Beringer et al., 2001; Gornall et al., 2007). In contrast to all other soil attributes, variation in depth of the active soil layer was associated with variation in vascular LAI. This positive association is clearly driven by flood meadow and wet fen, which both have high LAI and deep active soil layer, and bare soil and lichen tundra, which have low LAI and shallow active layer, and may be a consequence of several reciprocal plant-soil interactions. First, deep active soil can provide
more nutrients for leaf production than shallow soil. Second, accumulation of SOM in areas of higher LAI may increase the overall depth of soil above bedrock and the shallow biologically active layer in bare soil and lichen tundra may partly be due to the closeness of bedrock. Finally, variation in moss biomass may have a role in this pattern, too. Moss biomass and LAI were weakly negatively correlated across all plots and among flood meadow, graminoid tundra, bog and the two types of fens this negative association is clear. This suggests that despite a non-significant plot-to-plot correlation between
moss mass and active layer depth, soil insulation by mosses may be one reason for the positive link between LAI and





active layer depth. Overall, even though the ability of mosses to bind carbon per unit biomass may not be more than one-third of the ability of vascular plants (Korrensalo et al., 2016), our field data support the view that through their high biomass, mosses have a major role in structuring and driving the functioning of Arctic soils.

**4.2 Detecting field variation using remote sensing data**

Our data show that remote sensing reflectance data, and the NDVI index in particular, could far better capture variation in vascular LAI than moss biomass. This is not a new finding (Bratsch et al., 2017; Liu et al., 2017; Macander et al., 2017), but suggests difficulties in capturing soil variation using NDVI as many soil attributes at our site were linked to moss biomass. Indeed, although variation in soil attributes could be statistically significantly explained by variation in NDVI, the soil-NDVI relationships were mostly based on two groups of values, representing the barren and more vegetated sites, and NDVI could not satisfactorily capture variation within the more vegetated areas. Earlier studies have

shown that NDVI can capture variation in moss chlorophyll concentrations when the surface reflectance of field samples is measured in a laboratory (Lovelock and Robinson, 2002) as well as in moss layer thickness and moss photosynthesis in the field when moss areal cover approaches 100 % (Douma et al., 2007). In those sub-plots of our field site, where *Sphagnum* and other mosses were found (35 and 67 plots, respectively), their mean areal cover was 41 % and 36 % and median cover 20 % and 30 %, respectively. Apparently these percentages of cover are too low to produce a detectable

signal for effective remote sensing of moss abundance.

As suggested by earlier investigations (Langford et al., 2016; Juutinen et al., 2017), the ability of NDVI to capture variation in vegetation depended a lot on the timing of the satellite image. Variation in moss biomass was better captured by early-season images, most likely due to the low cover of vascular plant leaf area at that time, whereas late-season

images were needed to capture variation in peak LAI. Interestingly, due to this pattern, timing of the image affected the capture of soil variation as well. Variation in litter layer depth and soil temperature, both closely associated with moss biomass, were better captured by the early-season image, whereas variation in active layer depth, associated with vascular LAI, was better captured by the late-season image. These results demonstrate how multitemporal remote sensing data are essential for capturing the spatial variation of vegetation and soil in landscapes, where LCTs differ widely in plant

phenology. NDVI was recently found to be positively linked to active layer depth also in the Alaskan permafrost tundra by Gangodagamage et al. (2014). In their study, drier areas of thinner active layer were covered by lichen, mosses and dwarf shrubs and wetter areas of deeper active layer by mosses and *Carex*. This resembles our case and suggests that LAI mediated the positive association between NDVI and active layer depth in their study as well.

Contrary to our expectations and earlier findings (Suvanto et al., 2014; Emmerton et al., 2016; Riihimäki et al.,

2017), topographical features could not enhance the capture of spatial variation in plant and soil properties above the level achieved by NDVI when the timing of the satellite image was appropriate for the examined attribute (e.g. late-season image for capturing variation in LAI). However, in cases where the image timing was not optimal (e.g. late-season image for capturing variation in soil temperature), including relevant topographical attributes had an influence. Topography had many logical links to plant and soil attributes; e.g. elevation correlated negatively with vascular LAI, SOM content and

soil moisture, most likely due to the high plant and SOM production in low-land areas covered by wet fens. However, in contrast to what we anticipated, moss biomass was not significantly linked to any attribute of topography, and this is a likely reason why topography was not beneficial in our site: LAI was related to topography, but already well captured by reflectance, whereas moss biomass was neither related to topography nor satisfactorily captured by reflectance. One



potential reason for the lack of link between topography and moss biomass is our small study area, where moss growth can respond to microtopography only.

### 4.3 Extrapolating plant production and soil attributes in the landscape

As mosses formed a major part of vegetation, but could not adequately be captured by either NDVI or DEM, and as spatial variation in many soil parameters in our site was linked to variation in moss biomass, we conclude that the most feasible way to map spatial variation in vegetation and the soil in our landscape is to use plant community-based land cover classification maps. Variation in plant and soil parameters was relatively well explained by LCTs, which could then be classified from remotely sensed data. As expected, the relative importance of different LCTs in plant and SOM production did not follow their relative areal coverage in the landscape: shrub tundra exceeded its areal position in vascular shoot mass production (mostly due to heavy, woody parts of shrub biomass), wet fen in leaf area and SOM production, and dry fen and bog in moss and SOM production, while bare soil and lichen tundra had little significance in relation to their areal cover. These patterns seemed to remain through the growing season although soil active layer depth had different dynamics in different LCTs. It should be noted though that although we used multitemporal imagery and high-resolution DEM in constructing the land cover map, our classification accuracy remained low among those LCTs that had similar composition of plant functional types. This agrees well with our finding that variation within LCTs often overlapped each other and that not all LCTs differed statistically significantly from each other when analysed on the basis of functional parameters. Earlier studies have shown that if LCTs mostly differ in soil and bottom layer plant community composition, classification accuracies can be low (Davidson et al., 2016; Reese et al. 2014) and hyperspectral imagery may be needed for detecting differences among LCTs (Bratsch et al., 2016; Davidson et al., 2016; Liu et al., 2017). On the other hand, those LCTs that were most difficult to distinguish in our satellite image classification were those that were least different from each other in field measurements and multivariate data analysis. This suggests that the error that originates from the low classification accuracy when extrapolating plant and soil parameters in our tundra landscape is likely to be small.

*Data availability.* Once the manuscript is accepted for publication, the data will be archived to a data repository DRYAD (http://datadryad.org).

*Author contribution.* MA and TL established the study site. TV, ML, JM, EV, JN and MA designed field sampling and collected and analysed the samples. OP analysed soil OM% and VK contributed to field work. JM and DJK carried out statistical analysis of data. AR and TV processed the satellite images and produced the reflectance indices, DEM and land cover map. JM composed the manuscript with contributions from TV, ML, AR, DJK, TL, SJ and MA.

*Conflict of interest.* The authors declare that they have no conflict of interest.

*Special issue statement.* This article is part of the special issue "Changing Permafrost in the Arctic and its Global Effects in the 21st Century (PAGE21)"

*Acknowledgements.* We thank L. Rosenius for field and laboratory assistance, G. Chumachenko and O. Dmitrieva for kindly arranging our stay at the Tiksi Observatory, and Yakutian Service for Hydrometeorology and Environmental Monitoring for providing accommodation and access to the observatory. The study was financially supported by the Academy of Finland projects COUP (#291736), "Greenhouse gas, aerosol and albedo variations in the changing Arctic" (#269095) and "CAPTURE: Carbon dynamics across Arctic landscape gradients: past, present and future" (#296888 and #296423), Finnish Center of Excellence program (#272041) and EU FP7 project "Changing Permafrost in the Arctic and its Global Effects in the 21st Century" (#282700).



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



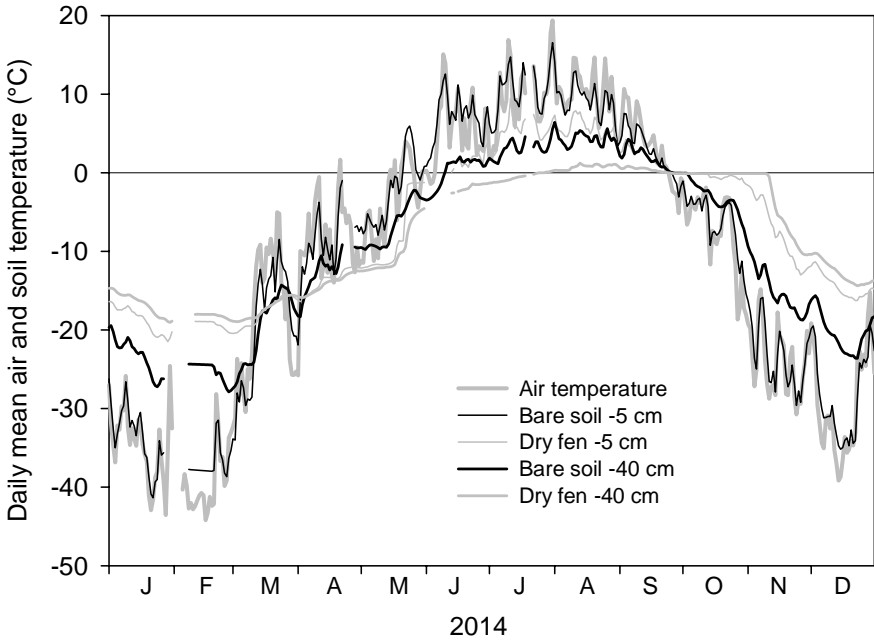

**Figure 1. Mean daily air temperature at the study area and the concomitant soil temperatures, measured 5 and 40 cm below ground surface at two field spots classified as bare soil and dry fen, in 2014.**



**Figure 2.** Land cover types of the study area: (a) bare soil with lichen tundra patches, (b) shrub tundra in the foreground, lichen tundra in the background, (c) bog, (d) mixture of dry and wet fen, (e) graminoid tundra, and (f) stream and flood meadow.





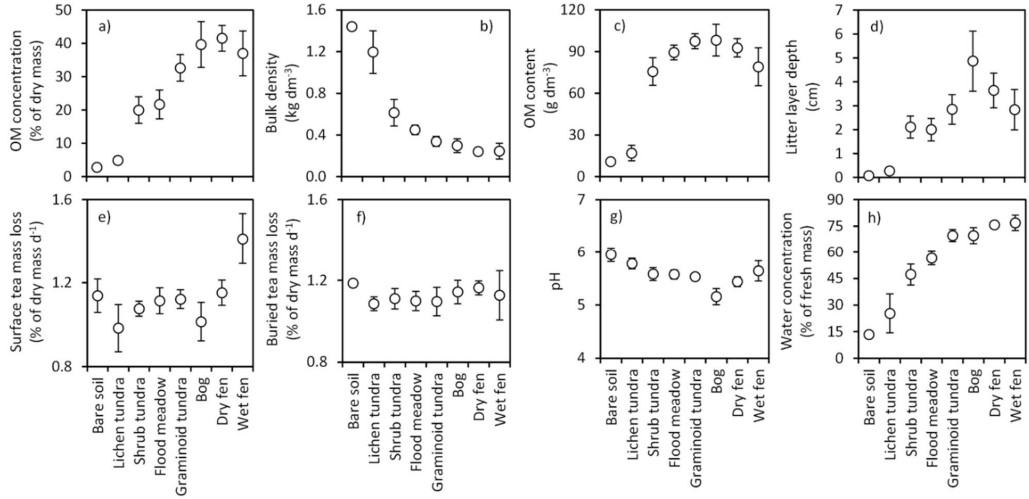

**Figure 3.** Means (± 85% CI) of (a) OM concentration, (b) bulk density and (c) OM content of the top 10 cm soil layer; (d) depth of the litter layer (including both vascular and moss plant material); mass loss of tea, (e) placed on the soil surface or (f) buried in the soil at a depth of 5 cm; and (g) pH and (h) water concentration of the top 10 cm soil layer in the land cover types of the Siberian Arctic tundra at Tiksi (arranged in order of increasing soil water concentration).

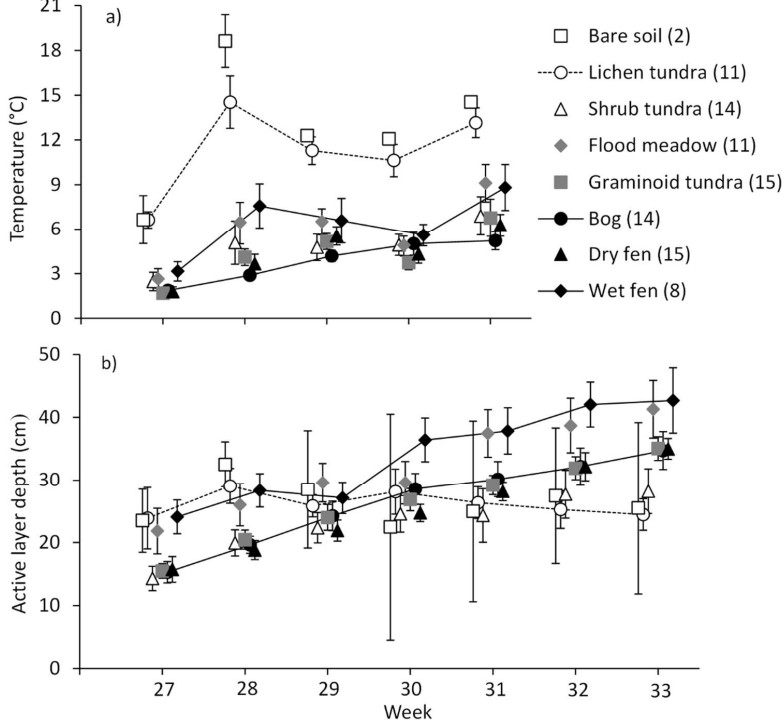

**Figure 4.** Development of (a) soil temperature (at a depth of 15 cm) and (b) depth of the active soil layer (mean ± 85% CI) during the growing season (week 27 represents early July with 160DD, week 31 early August with 380 DD and week 33 mid-August with 550 DD) in the land cover types of Tiksi tundra (the number of replicate plots is presented in brackets; for the sake of clarity, land cover types are deviated from each other within weeks and only some have means connected with lines).





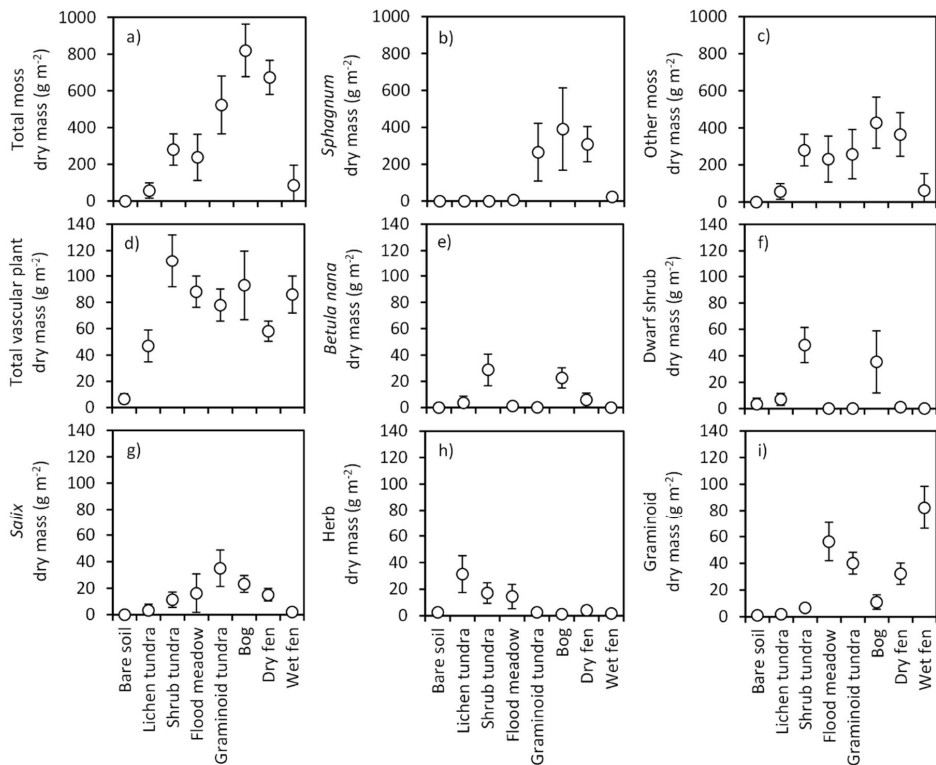

**Figure 5. Biomass (mean ± 85% CI) of (a) all mosses, (b) *Sphagnum*, (c) mosses excluding *Sphagnum*, (d) all vascular plants, (e) *Betula nana*, (f) dwarf shrubs, (g) *Salix*, (h) herbs and (i) graminoids in late July (ca. 360 DD) in the land cover types of Tiksi tundra (arranged in order of increasing soil water concentration).**

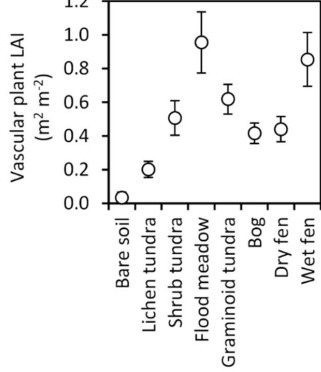

**Figure 6. Leaf area index (LAI, mean ± 85% CI) of vascular plants in late July (ca. 360 DD) in the land cover types of Tiksi tundra (arranged in order of increasing soil water concentration).**





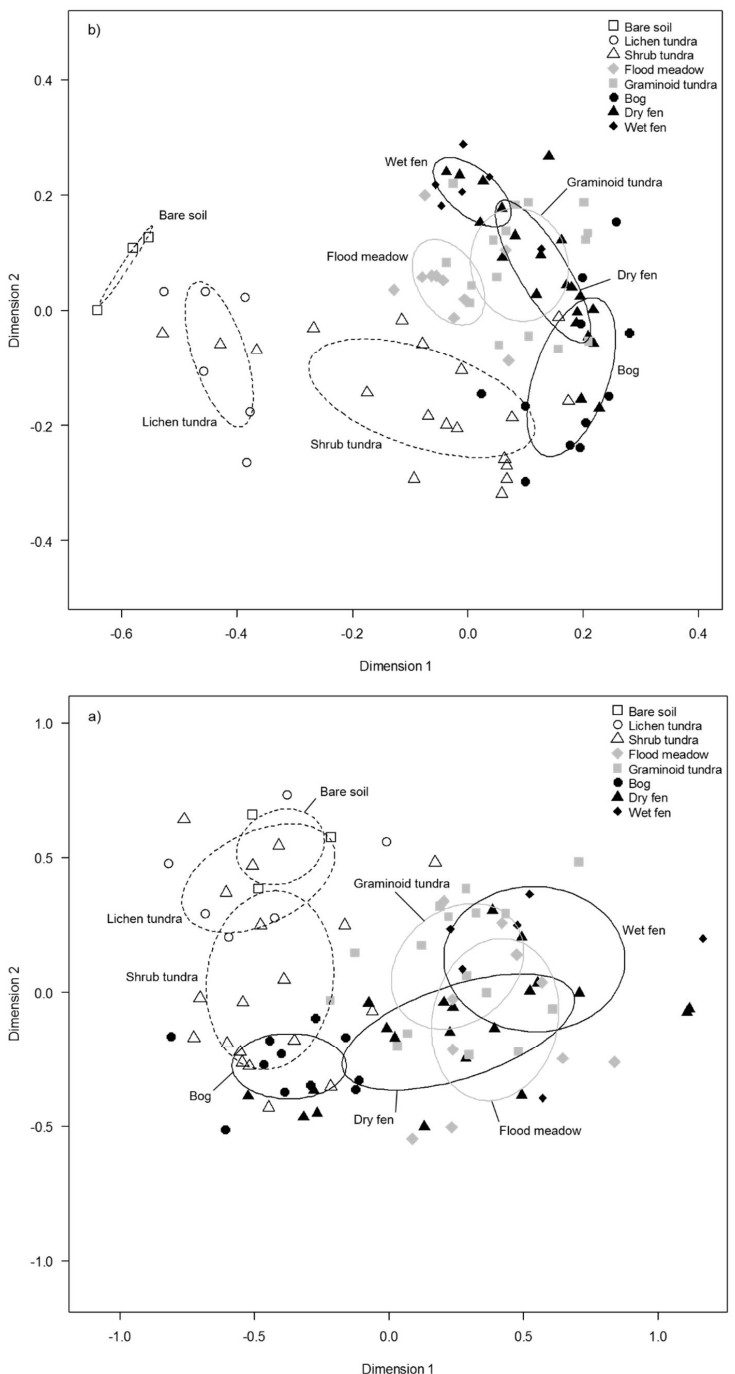

**Figure 7.** Non-metric multidimensional scaling (NMDS) graphs of **(a)** dicotyledonous plant species (presence/absence data) and **(b)** the combined data of plant functional group biomasses and soil variables in ordination planes with the eight land cover types (LCTs), judged visually at the field site, as an overlay (dispersion ellipses indicate 1 SD of the weighted averages of LCT scores).



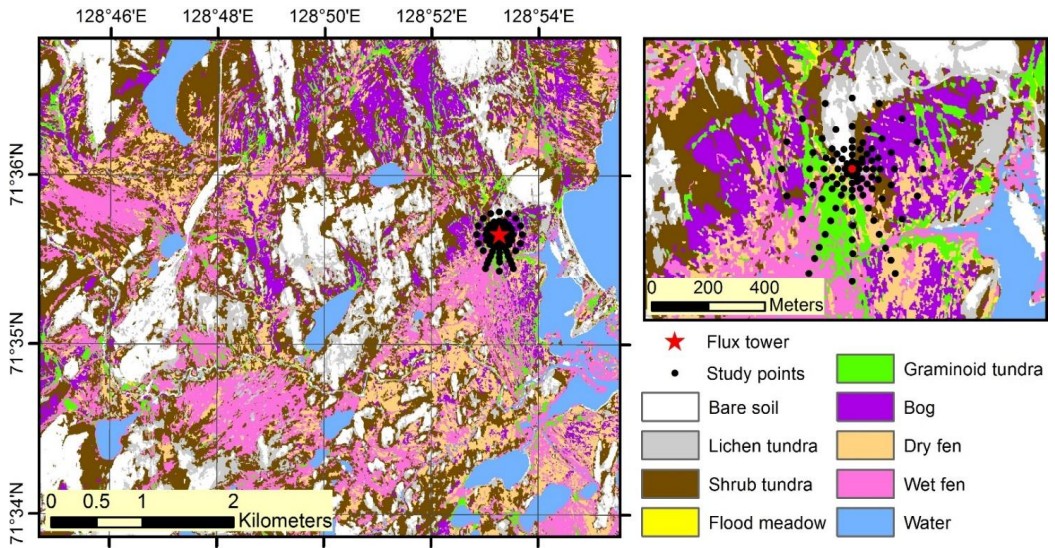

**Figure 8. Land cover map of the study landscape.**



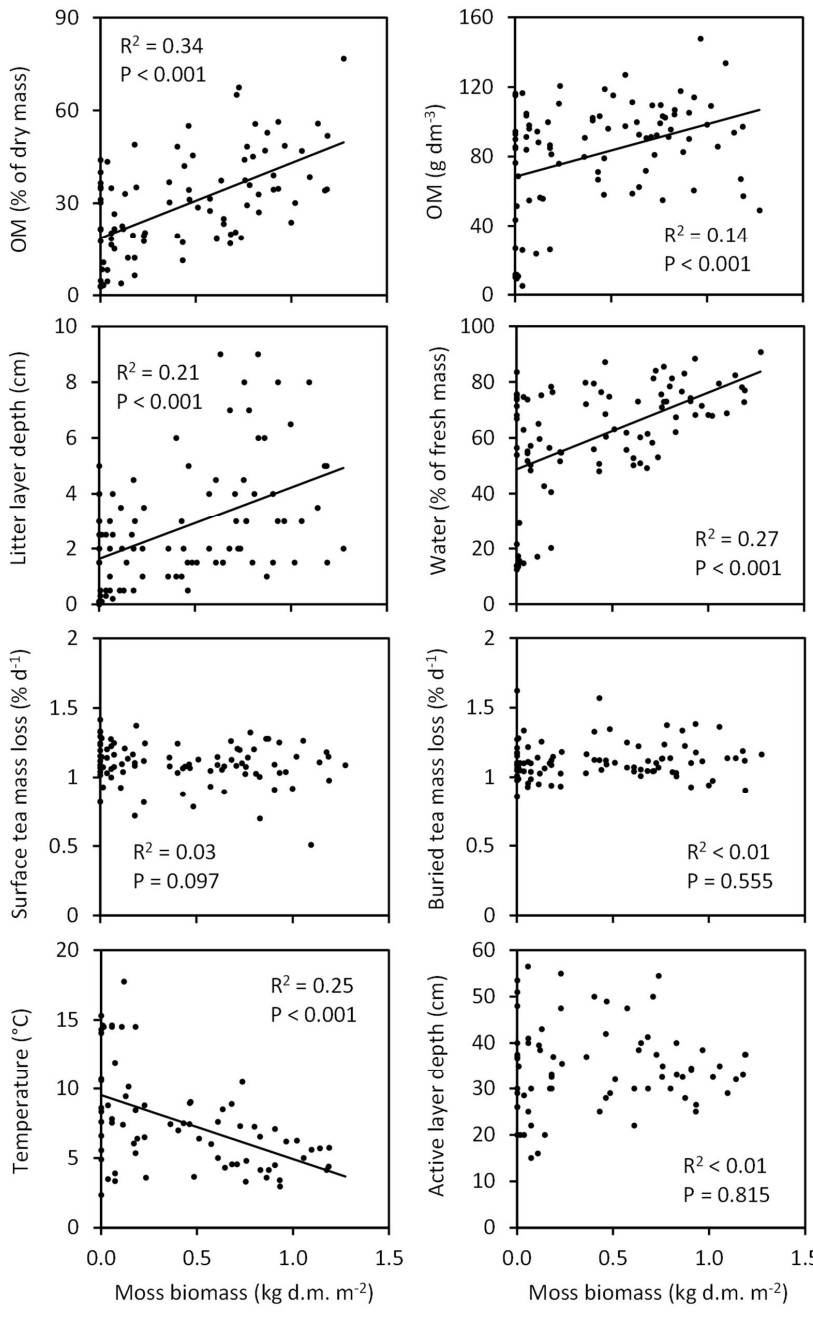

**Figure 9. Associations of moss biomass with soil characteristics across the Tiksi tundra field plots ($R^2$ and P values are from linear regression analyses, lines are shown for statistically significant associations only; n = 92 except for temperature and active layer depth, where n = 73; soil OM%, OM content and moisture are for the top 10 cm soil layer, litter layer consists of both vascular and moss plant material, teabags were buried and temperature measured at a depth of 5 and 15 cm, respectively; temperature and active layer depth represent week 31 and 33 measurements, respectively).**





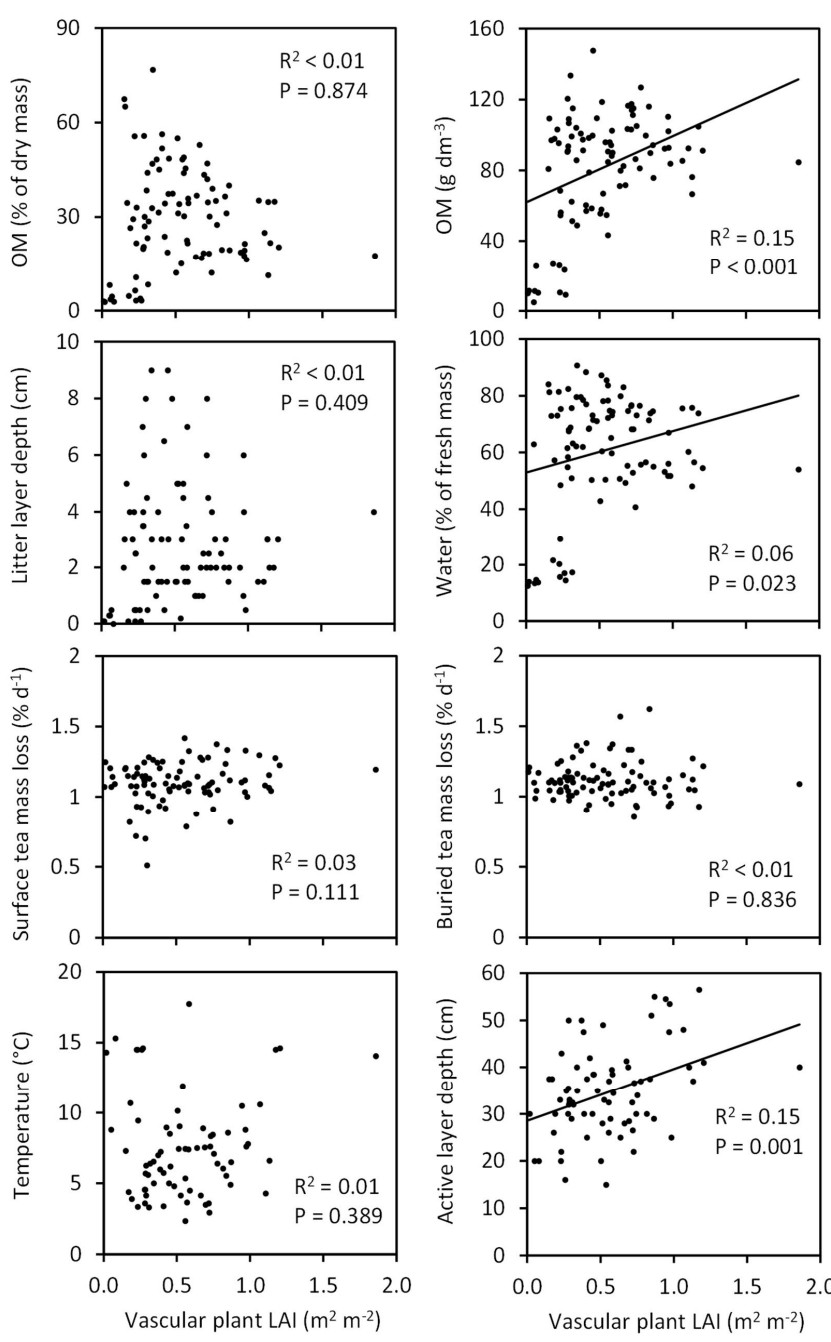

**Figure 10. Associations of vascular plant leaf area index (LAI, measured in late July with ca. 360 DD) with soil characteristics across the Tiksi tundra field plots (see Fig. 9 for an explanation of the data and graphs).**



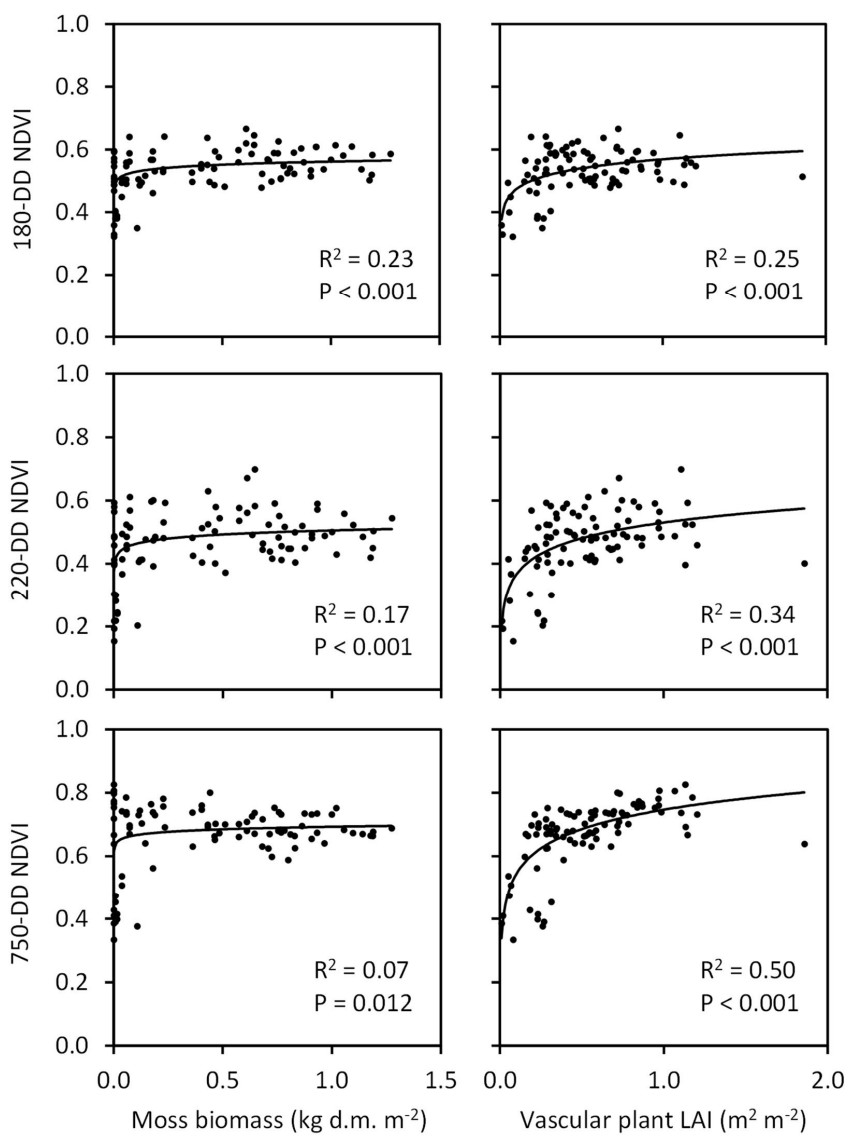

**Figure 11. Associations of moss biomass and vascular plant leaf area index (LAI, measured in late July at ca. 360 DD) with NDVI extracted from QB (taken at 180 DD) and WV-2 images (taken at 220 and 750 DD) across the Tiksi tundra field plots (n = 92, $R^2$ and P values are from logarithmic regression analysis).**





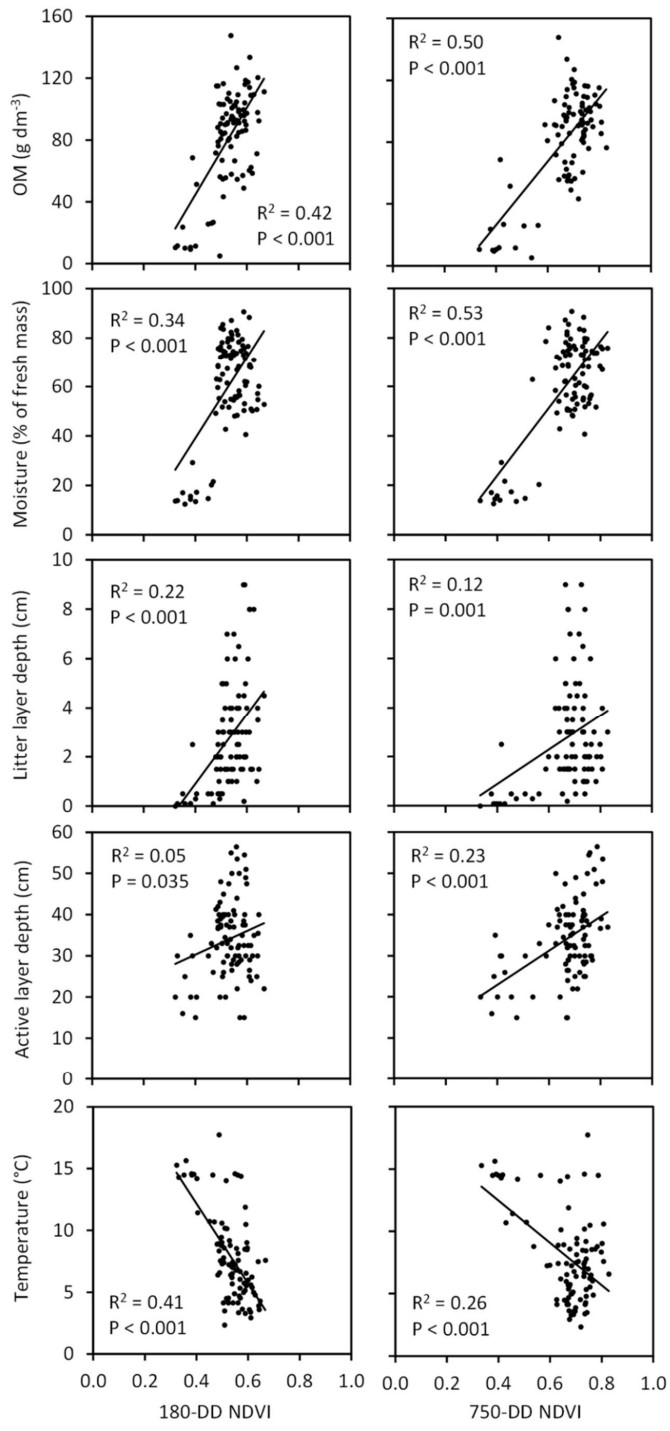

**Figure 12. Associations of NDVI, extracted from QB (taken at 180 DD) and WV-2 (taken at 750 DD) images, with soil characteristics across the Tiksi tundra field plots ($R^2$ and P values, regression lines and soil variables are as in Fig. 9, except that n = 92 in all graphs).**



**Table 1. Criteria used in the field to visually distinguish the land cover types of the study area (the final, applied types are in bold).**

| Land cover type | Description |
| --- | --- |
| Peatlands | Noticeable peat layer. Peat forming plants (*Sphagnum*, *Carex*, *Eriophorum*) and shrubs. Split here into fens and bog. |
| Fens | Wetter peatlands. *Carex* and brown mosses dominate. Split further into dry and wet fen. |
| **Dry fen** | Water surface below the moss layer. Some shrubs may occur. |
| **Wet fen** | Water table high, often water pools. Mainly *Carex*, some mosses. |
| **Bog** | Drier peatlands, hummock-hollow patterns. Dwarf shrubs and *Betula nana* common. *Sphagnum* dominates the moss layer. |
| Moorlands/Heaths | Dry areas, thin humus layer, no peat formation, mineral soil close to the soil surface. Shrubs dominate, but also annuals, grasses, heath mosses, lichens, no *Sphagnum*. Split here into tundra heaths and graminoid tundra. |
| Tundra heaths | Lichen or shrub dominated. Split further into lichen and shrub tundra. |
| **Lichen tundra** | Lichen dominated, but also a few dwarf shrubs, annuals and mosses, no *Sphagnum*. Often in patches surrounded by bare ground. |
| **Shrub tundra** | Shrub dominated, but also lichens, annuals and mosses, no *Sphagnum*. |
| **Graminoid tundra** | Grass dominated areas. *Salix*, shrubs, annuals and other vascular plants (e.g. *Polytrichum*, *Dicranum*) may occur. |
| Meadows | Riverside spring flooding areas, drier during growing season. |
| **Flood meadow** | Grass dominated, *Salix* common, annuals occur, brown mosses, no *Sphagnum*. |
| Non-vegetated | |
| **Bare soil** | Stony, non-vegetated areas. |
| **Water** | |





**Table 2. The number (*n*) and mean distance of study plots (within-type delta) within land cover types (LCTs), and the *T*-statistic and *P*-values of the mean distance of LCTs when the analysis is based on the presence/absence data of dicotyledonous plants (lower-left triangle) or plant functional group biomasses and soil parameters (upper-right triangle). Significant P-values are highlighted in bold.**

| Land cover type | | | Bare soil | Lichen tundra | Shrub tundra | Flood meadow | Graminoid tundra | Bog | Dry Fen | Wet fen |
|---|---|---|---|---|---|---|---|---|---|---|
| | | *n* | 3 | 6 | 19 | 10 | 16 | 11 | 21 | 6 |
| | Within-type delta | | 0.127 | 0.529 | 0.497 | 0.394 | 0.539 | 0.475 | 0.462 | 0.292 |
| Bare soil | 0.018 | *T* | | -3.52 | -6.44 | -6.89 | -7.27 | -6.41 | -8.55 | -5.11 |
| | | *P* | | **0.014** | **<0.001** | **0.004** | **0.001** | **0.003** | **<0.001** | **0.011** |
| Lichen tundra | 0.059 | *T* | 0.44 | | -3.93 | -6.64 | -7.40 | -7.32 | -9.64 | -6.16 |
| | | *P* | 0.628 | | **0.007** | **<0.001** | **<0.001** | **<0.001** | **<0.001** | **0.002** |
| Shrub tundra | 0.147 | *T* | -2.58 | -2.89 | | -3.26 | -6.39 | -4.71 | -10.24 | -7.07 |
| | | *P* | **0.024** | **0.014** | | **0.014** | **<0.001** | **0.002** | **<0.001** | **<0.001** |
| Flood meadow | 0.148 | *T* | -5.32 | -7.61 | -11.46 | | -1.73 | -5.08 | -6.07 | -2.35 |
| | | *P* | **0.003** | **<0.001** | **<0.001** | | 0.065 | **<0.001** | **<0.001** | **0.038** |
| Graminoid tundra | 0.133 | *T* | -4.94 | -8.58 | -13.98 | -2.69 | | -1.39 | -0.95 | -2.68 |
| | | *P* | **0.002** | **<0.001** | **<0.001** | **0.018** | | 0.094 | 0.140 | **0.019** |
| Bog | 0.040 | *T* | -5.80 | -7.67 | -4.48 | -9.93 | -10.36 | | -0.76 | -6.43 |
| | | *P* | **0.003** | **<0.001** | **0.002** | **<0.001** | **<0.001** | | 0.166 | **<0.001** |
| Dry Fen | 0.192 | *T* | -5.36 | -8.85 | -14.32 | -4.74 | -2.12 | -7.77 | | -5.56 |
| | | *P* | **<0.001** | **<0.001** | **<0.001** | **0.001** | **0.036** | **<0.001** | | **<0.001** |
| Wet fen | 0.194 | *T* | -4.12 | -5.96 | -9.85 | -2.83 | -1.61 | -8.09 | -1.07 | |
| | | *P* | **0.013** | **0.002** | **<0.001** | **0.007** | 0.069 | **<0.001** | 0.130 | |





**Table 3. Land cover types, their areal cover (water covered 9.38 %) within the 35.8 km$^2$ landscape (Fig. 8), and the quantity of leaf area, plant biomass and biologically active OM (in the unfrozen soil layer) in each type (% of grand total given in brackets), measured during the early (160 DD), mid (350 DD) and late (550 DD) growing season.**

| Land cover type | Areal cover (%) | Peak season vascular leaf area (km$^2$) | Peak season vascular shoot mass (Gg) | Peak season moss biomass (Gg) | Early season SOM (Gg) | Late season SOM (Gg) |
|---|---|---|---|---|---|---|
| Shrub tundra | 26.21 | 4.76 (33) | 1.05 (44) | 2.63 (29) | 102 (28) | 201 (29) |
| Wet fen | 15.64 | 4.78 (33) | 0.48 (20) | 0.48 (5) | 107 (29) | 189 (27) |
| Bare soil | 14.63 | 0.18 (1) | 0.04 (2) | 0 (0) | 13 (4) | 14 (2) |
| Dry fen | 11.13 | 1.76 (12) | 0.23 (10) | 2.69 (29) | 58 (16) | 129 (18) |
| Lichen tundra | 10.66 | 0.77 (5) | 0.18 (8) | 0.22 (2) | 16 (4) | 16 (2) |
| Bog | 8.68 | 1.29 (9) | 0.29 (12) | 2.55 (28) | 47 (13) | 107 (15) |
| Graminoid tundra | 3.24 | 0.72 (5) | 0.09 (4) | 0.61 (7) | 18 (5) | 41 (6) |
| Flood meadow | 0.42 | 0.14 (1) | 0.01 (0.5) | 0.04 (0.4) | 3 (1) | 6 (1) |
| Grand total in region | | 14.4 | 2.37 | 9.22 | 364 | 703 |





**Table 4. Coefficients of determination ($R^2$) of regression models that included those topographical features that correlated statistically significantly ($P < 0.05$) with the dependent variable, the best or worst NDVI predictor (Figs. 11 and 12) and the best or worst NDVI predictor amended with the topographic features.**

| Dependent variable | Correlating features of topography | Topography only | Best NDVI | Best NDVI + topography | Worst NDVI | Worst NDVI + topography |
|---|---|---|---|---|---|---|
| Vascular plant LAI (log) | Elevation, TWI, TPI-100 | 0.22 | 0.50 | 0.52 | 0.25 | 0.37 |
| Soil OM content | Elevation | 0.16 | 0.50 | 0.51 | 0.42 | 0.48 |
| Soil moisture | Elevation | 0.19 | 0.53 | 0.55 | 0.34 | 0.44 |
| Litter layer depth | Elevation, SR | 0.07 | 0.22 | 0.24 | 0.12 | 0.15 |
| Active layer depth | Elevation, slope, SR | 0.15 | 0.23 | 0.31 | 0.05 | 0.18 |
| Soil temperature | Slope, TPI-25 | 0.10 | 0.41 | 0.46 | 0.26 | 0.34 |

