# Peer review of "Spatial variation and linkages of soil and vegetation in the Siberian Arctic tundra – coupling field observations with remote sensing data"

_Biogeosciences, 2017_

## Referee Comment (RC1) · Anonymous Referee #1 · 6 Feb 2018

The presented manuscript analyses spatial variation of plant and soil properties and their relations to each other for a field site in the Siberian Arctic tundra. Furthermore, it is tested to what extent remote sensing data can be utilised to capture variation in these properties and, consequently, to extrapolate vegetation and soil effects on ecosystem carbon fluxes to the large scale. The study highlights difficulties in predicting soil properties from NDVI, since they are not linked to vascular plant LAI, but moss biomass, which cannot be captured well by remote sensing. Instead, a classification of vegetation and soil properties according to land cover types is recommended to capture their spatial variation. The manuscript is well written and of good scientific quality. However, some details of the methodology and the results should be explained more clearly in

order to make the manuscript easier to understand for readers who are not familiar with remote sensing techniques. I recommend to publish it with minor revisions, as outlined below.

General comments

(1) The abstract is too long in my opinion, the middle part contains many details and figures which are not essential for the message of the study. I therefore recommend to shorten it by around 50%.

(2) A schematic of the individual working steps described in the methods section and how they relate to each other may be helpful to understand the approach (e.g. how are data from field sampling combined with remote sensing),

(3) As far as I understood Sect. 2.5, the 9 land cover types (LCTs) were determined through personal judgement in the field. Subsequently, a statistical classification approach (random forest) was applied to construct a model which predicts these LCTs based on remote sensing data. The authors report that 109 features of the remote sensing images were used in this model to predict the LCTs. However, no information is provided on what exactly these features are and why such a large number of input variables is needed. Please explain (a) what properties these 109 features represent (in general terms). (b) how you tested the model for overfitting, i.e. couldn't you have produced a similar prediction of LCTs with less input variables? (I am aware that the number of original features was reduced from 262 to 109, but this still seems a lot to me) (c) why the features lead to a superior recognition of LCTs than the NDVI-based information.

(4) Figures 3 to 6 show how vegetation and soil properties depend on LCT, while Figs. 11 and 12 show relations between moss biomass/vascular LAI and NDVI, and soil properties and NDVI, respectively. While around half of the relations in Figs. 11 and 12 are relatively weak (R squared < 0.3), the others do suggest that NDVI provides information about vegetation and soil properties. Moreover, the LCT classification does

show some weak relations to soil and vegetation properties, too, and the external accuracy of the statistical model which predicts LCT is only 49%. Hence, it is not obvious to me why the derivation of vegetation and soil properties via LCTs is better than the NDVI-based derivation. It would be nice if you could provide a more quantitative prediction of soil and vegetation properties based on LCTs. Early growing season NDVI, for instance, explains 23% of moss biomass. Would it be possible to come up with a comparable figure for the LCT approach, e.g. given the 50% accuracy in predicting the LCT, and the standard deviation in moss biomass within an LCT, how much of the variance is explained?

(5) I am missing a few words on the outcome of the decomposition experiment in the discussion section. What is the implication for ecosystem carbon fluxes?

Specific comments

p3,l1 Could you please expand the sentence by one or two examples stating which soil properties affect ecosystem carbon exchange and how they do that?

p4,l25 Please add a few words on why this sampling point pattern was chosen (increasing distances between points with larger distance from EC tower). In particular, explain how this pattern is suitable to capture soil and vegetation properties at different spatial scales (from smaller to larger distances) for the study area.

p5,l4 Please explain shortly why these soil properties were measured? How do they relate to carbon exchange fluxes?

p10,l30 & p11,l12 & p14,l1 Shouldn't moss biomass relate to topography via wetness? At least Sphagnum should show a link to low elevation. Figure 8 seems to show a good correlation between topography and dry/wet areas, which correlate well with vegetation type. Therefore, the explanation regarding microtopography seens not very satisfactory to me. Please explain this in more detail and maybe show a map of the topography of the study area for comparison.

p11,l15f If remote sensing reflectance cannot capture well moss biomass and associated soil properties, how can remote sensing be successfully used to classify LCTs, which also largely depend on vegetation properties? This means, how can the random forest classification distinguish between the 7 LCTs which differ mostly in vegetation properties? Please provide more details on the 262/109 features (see also general comment 3 & 4).

Comments on style

p4,l13 "..the soil is in continuous.." - the "in" seems to be misplaced here.

p4,l20 What do you mean by the word "manuscript" in the cited studies? At least Tuovinen et al do not appear in the bibliography. Could you please correct that and use "submitted" instead?

p6,l12 Please explain the abbreviation "GCP", e.g. putting it in brackets in line 10.

p9,l29 It is not clear what "user and producer accuracies" are.

p13,l10 Please provide a reference to the figure/table which illustrate this finding at the end of the sentence. This should be done also for the rest of the discussion.

---

## Referee Comment (RC2) · Anonymous Referee #2 · 7 Feb 2018

Juha Mikola and colleagues present a study from northern Siberia that focused on (1) spatial variation in plant and soil attributes within a tundra ecosystem, (2) co-variation in these attributes, and (3) the potential to map these attributes using remote sensing. The researchers show that plant and soil attributes (e.g., plant biomass, soil organic matter content) differed among land cover types and that both moss biomass and vascular plant leaf area index (LAI) were weakly to moderately correlated with several soil attributes. Furthermore, they examined whether the plant and soil attributes could be mapped using the normalized difference vegetation index (NDVI) derived from very high spatial resolution satellite imagery that was acquired at different points during the growing seasons. This comparison showed that moss biomass was most closely

related to early summer NDVI, whereas vascular plant LAI was more closely related to mid-summer NDVI, which suggests that multi-temporal imagery may be useful for quantifying different aspects of plant and soil attributes in tundra ecosystems. The researchers conclude that spatial extrapolation of plant and soil attributes may require the use of land cover maps and field sampling within land cover types rather than linking field measurements directly with remote-sensing observations. In general, the study is robust and multi-faceted, and the manuscript is very well written. Overall, the study makes a valuable contribution to arctic ecology and would be well-suited for Biogeosciences, though could benefit from some refinements detailed below.

General comments

(i) I agree with reviewer 1 that the abstract is too long and detailed. The primary findings and implications would be better highlighted if the abstract was condensed.

(ii) One of the primary conclusions from this study is that spatial extrapolation of plant and soil attributes will require using land cover maps and field sampling within different land cover types. This approach contrasts with the direct remote sensing approach that involves (1) developing statistical relationship between field and surface reflectance measurements and then (2) modeling plant/soil attributes across a broader area by applying these statistical relationships to wall-to-wall surface reflectance measurements. The authors' conclusion is based on the observation that NDVI often explained little of the spatial variation in plant and soil attributes across the network of plots. This comparison alone does not seem like an adequate basis from which to draw the conclusion stated above. A direct remote sensing approach does not need to be based solely on NDVI, but rather plant/soil attributes could be predicted using all the spectral bands and plus derived texture metrics and spectral indices. Further analysis might support the authors' current conclusions, but the current conclusion seems premature given the analysis presented.

(iii) The authors estimated the total amount of leaf area, plant biomass, and soil organic

mass that occurred within several land cover types found in their study area; however, these numbers do not include estimates of uncertainty. The lack of uncertainty estimates also fits with my comment above (ii). The authors could estimate uncertainty in these totals based on variation in attributes within each land cover type or could perhaps use a Monte Carlo approach in which they account for variation with each land cover type as well as uncertainty in the land cover map.

(iv) The tea bag decomposition measurements don't seem to fit with any of the stated objectives and are not mentioned in the discussion. The manuscript as already has quite a few elements, so I'd suggest dropping those measurements from the manuscript and focusing on the core elements.

(v) It would be good to note in the discussion that the remote sensing observations were not acquired concurrent with field sampling, which introduces uncertainty into the comparisons between NDVI and field measurements. The QuickBird imagery was acquired almost a decade prior to field sampling, whereas the WorldView-2 images were acquired with a year or two of field sampling. These time lags could make it harder to relate field measurements to NDVI, especially for the QuickBird imagery.

Specific comments

(i) P4, L10: Spell out "DD" the first time it is used.

(ii) P4, L37-39: Please clarify whether you harvested live vascular shoot biomass, or live + standing dead vascular shoot biomass. Also, please describe how you defined the bottom of the moss layer.

(iii) P5, L1: Define leaf area index (m2 leaf m-2 ground) and whether LAI was based on projected leaf area, hemi-leaf area, or two-sided leaf area. It seems you used projected leaf area.

(iv) P5, L26-27: I'd encourage the authors to put all of the plot-level measurements in the supplemental material, as well as the current summary for each land cover type.

The current summary table should also probably include the standard deviation of each measurement for each land cover type.

(v) P6, L5: You might add that NDVI has been used not only for "spatial examination of LAI" but also for mapping plant aboveground biomass in tundra ecosystem (e.g., Raynolds et al. 2012, Berner et al. 2018). Raynolds: http://www.tandfonline.com/doi/abs/10.1080/01431161.2011.609188 Berner: http://iopscience.iop.org/article/10.1088/1748-9326/aaaa9a

(vi) P6, L8: Technically, you generated a digital surface model (DSM) rather than a digital elevation model (DEM). A DSM includes the height of vegetation and other features, while a DEM represents bare-year elevation.

(vii) P6, L10: Define "ground control point" acronym (GCP) in this sentence, which is the first time the term is mentioned.

(viii) P 13, L33-38: What is the range in elevation among the field plots? It is probably quite small. Could it be that topography wasn't a strong predictor of plant/soil attributes because the digital surface model was not accurate enough to differentiate small, but ecologically important differences in elevation among plots? Topography might be a stronger predictor where there is greater topographic variation among field plots.

Tables and figures

F1. It is hard for me to differentiate some of the lines used in the figure. Could these be plotted using color, or a variety of line types?

F3. Spell out "OM" in the figure legend before using the abbreviation.

F4. Specify that the plotting symbols represent averages and error bars represent 85% CI.

F7. Increase the size of text in the figure.

F8. Proved a little more detail in the legend, such as how the land cover map was

derived.

F9. Most of the legend is contained with in parentheses and separated by a bunch of semi-colons and comma. Breaking those five lines into several sentences would make it easier to read.

T2. Perhaps note in the table caption that these numbers are derived using the multi-response permutation procedure.

T3. The column names "early season SOM" and "late season SOM" are somewhat confusing. Maybe it would be clearer to label those two columns as "SOM in unfrozen soil (Gg)" and then have sub-column names labeled "Early season" and "Late season".

---

## Author Comment (AC1) · 16 Mar 2018

Thank you for your very helpful comments. We have revised our manuscript accordingly. Our responses to your comments are listed below one by one. The page and line numbers refer to the revised manuscript, which we provide as a supplement to this response.

......

Referee #1 comments

The presented manuscript analyses spatial variation of plant and soil properties and

their relations to each other for a field site in the Siberian Arctic tundra. Furthermore, it is tested to what extent remote sensing data can be utilised to capture variation in these properties and, consequently, to extrapolate vegetation and soil effects on ecosystem carbon fluxes to the large scale. The study highlights difficulties in predicting soil properties from NDVI, since they are not linked to vascular plant LAI, but moss biomass, which cannot be captured well by remote sensing. Instead, a classification of vegetation and soil properties according to land cover types is recommended to capture their spatial variation. The manuscript is well written and of good scientific quality. However, some details of the methodology and the results should be explained more clearly in order to make the manuscript easier to understand for readers who are not familiar with remote sensing techniques. I recommend to publish it with minor revisions, as outlined below.

General comments (1) The abstract is too long in my opinion, the middle part contains many details and figures which are not essential for the message of the study. I therefore recommend to shorten it by around 50%. REPLY#1 – We shortened the abstract.

(2) A schematic of the individual working steps described in the methods section and how they relate to each other may be helpful to understand the approach (e.g. how are data from field sampling combined with remote sensing). REPLY#2 – We added a diagram of the main working steps (Figure 2).

(3) As far as I understood Sect. 2.5, the 9 land cover types (LCTs) were determined through personal judgement in the field. Subsequently, a statistical classification approach (random forest) was applied to construct a model which predicts these LCTs based on remote sensing data. The authors report that 109 features of the remote sensing images were used in this model to predict the LCTs. However, no information is provided on what exactly these features are and why such a large number of input variables is needed. Please explain (a) what properties these 109 features represent (in general terms). (b) how you tested the model for overfitting, i.e. couldn't you have
produced a similar prediction of LCTs with less input variables? (I am aware that the number of original features was reduced from 262 to 109, but this still seems a lot to me) (c) why the features lead to a superior recognition of LCTs than the NDVI-based information. REPLY#3 – (a) Previous studies have shown that inclusion of multiple features improves classification accuracies. For this reason, we used several features, calculated from different datasets, to capture topographic and spectral variation. We earlier explained in general terms what these features were, but now we also included more justification for why we used so many (P7, L3-5). (b) Random forest is insensitive to overfitting and handles well multidimensional data. Therefore, the high number of features is not a problem. However, it has earlier been shown that feature selection improves the performance of random forest. We included this reasoning in the manuscript (P7, L16-18). (c) The features included NDVI, but also captured other aspects. This ultimately leads to better recognition of LCTs than with NDVI alone.

(4) Figures 3 to 6 show how vegetation and soil properties depend on LCT, while Figs. 11 and 12 show relations between moss biomass/vascular LAI and NDVI, and soil properties and NDVI, respectively. While around half of the relations in Figs. 11 and 12 are relatively weak (R squared < 0.3), the others do suggest that NDVI provides information about vegetation and soil properties. Moreover, the LCT classification does show some weak relations to soil and vegetation properties, too, and the external accuracy of the statistical model which predicts LCT is only 49%. Hence, it is not obvious to me why the derivation of vegetation and soil properties via LCTs is better than the NDVI-based derivation. It would be nice if you could provide a more quantitative prediction of soil and vegetation properties based on LCTs. Early growing season NDVI, for instance, explains 23% of moss biomass. Would it be possible to come up with a comparable figure for the LCT approach, e.g. given the 50% accuracy in predicting the LCT, and the standard deviation in moss biomass within an LCT, how much of the variance is explained? REPLY#4 – This is a good point and we therefore calculated measures of uncertainty for the predictions of the LCT approach as well (P7, L36 – P8, L3; P10, L23-29; Table 3, see also our REPLY#18). These show that the uncertainty

in capturing and predicting moss biomass cannot be avoided in LCT map either (P12, L3-8; P15, L23-27). We revised the discussion and conclusions based on these new results.

(5) I am missing a few words on the outcome of the decomposition experiment in the discussion section. What is the implication for ecosystem carbon fluxes? REPLY#5 - We now better illustrate the implication of the results of the tea bag trial for ecosystem carbon fluxes (P13, L15-28).

Specific comments p3,l1 Could you please expand the sentence by one or two examples stating which soil properties affect ecosystem carbon exchange and how they do that? REPLY#6 – We added temperature as an example of important soil properties (P2, L34-35).

p4,l25 Please add a few words on why this sampling point pattern was chosen (increasing distances between points with larger distance from EC tower). In particular, explain how this pattern is suitable to capture soil and vegetation properties at different spatial scales (from smaller to larger distances) for the study area. REPLY#7 – We added our reasoning for the chosen study plot pattern and contemplate its effects on capturing variation in soil and vegetation properties (P4, L18-21).

p5,l4 Please explain shortly why these soil properties were measured? How do they relate to carbon exchange fluxes? REPLY#8 – We added reasons for measuring the chosen soil properties (P5, L1-3).

p10,l30 & p11,l12 & p14,l1 Shouldn't moss biomass relate to topography via wetness? At least Sphagnum should show a link to low elevation. Figure 8 seems to show a good correlation between topography and dry/wet areas, which correlate well with vegetation type. Therefore, the explanation regarding microtopography seems not very satisfactory to me. Please explain this in more detail and maybe show a map of the topography of the study area for comparison. REPLY#9 – We revised and elaborated the discussion about the reasons for the lack of link between topography and moss

mass (P15, L4-11).

p11,l15f If remote sensing reflectance cannot capture well moss biomass and associated soil properties, how can remote sensing be successfully used to classify LCTs, which also largely depend on vegetation properties? This means, how can the random forest classification distinguish between the 7 LCTs which differ mostly in vegetation properties? Please provide more details on the 262/109 features (see also general comment 3 & 4). REPLY#10 – The RF classification was specifically trained to classify LCTs, i.e. we used training data of LCT occurrence when constructing the classification (P7, L14-16). In addition, in the RF classification, we used eight spectral bands and three spectral indices of two satellite images taken at different phases of the growing season as well as several topographic features. In the regressions, we used one spectral index and topographical features only. Nevertheless, although the larger set of features in LCT classification helped in capturing variation in vegetation, the overall classification accuracy was 49%, thus suggesting that there is uncertainty in the LCT classification as well (P15, L30-31). The 262 features consisted of 15 features calculated from spectral bands and 2 features calculated from spectral index and topographical layers. The 109 features is a subset of these 262 features and included spectral, topographic and textural features. These details are now better explained in the text (P7, L11-13 and 20-21).

Comments on style p4,l13 "..the soil is in continuous.." - the "in" seems to be misplaced here. REPLY#11 – Right, "in" was deleted.

p4,l20 What do you mean by the word "manuscript" in the cited studies? At least Tuovinen et al do not appear in the bibliography. Could you please correct that and use "submitted" instead? REPLY#12 – We removed citations to unpublished papers.

p6,l12 Please explain the abbreviation "GCP", e.g. putting it in brackets in line 10. REPLY#13 – Done.

p9,l29 It is not clear what "user and producer accuracies" are. REPLY#14 – We added

explanations for the two accuracies (P10, L10-11).

p13,l10 Please provide a reference to the figure/table which illustrate this finding at the end of the sentence. This should be done also for the rest of the discussion. RE-PLY#15 – To help the reader, we added references to Figures and Tables throughout discussion.

Please also note the supplement to this comment:
https://www.biogeosciences-discuss.net/bg-2017-569/bg-2017-569-AC1-supplement.pdf

---

## Author Comment (AC2) · 16 Mar 2018

Thank you very much for your helpful comments. We used them to revise our manuscript. Our responses to your comments are listed below one by one. Page and line numbers refer to the revised manuscript, which we provide as a supplement to this response. Response numbers continue the list from our responses to Referee 1.

....

Referee 2 comments

Juha Mikola and colleagues present a study from northern Siberia that focused on (1)

[Figure]

spatial variation in plant and soil attributes within a tundra ecosystem, (2) co-variation in these attributes, and (3) the potential to map these attributes using remote sensing. The researchers show that plant and soil attributes (e.g., plant biomass, soil organic matter content) differed among land cover types and that both moss biomass and vascular plant leaf area index (LAI) were weakly to moderately correlated with several soil attributes. Furthermore, they examined whether the plant and soil attributes could be mapped using the normalized difference vegetation index (NDVI) derived from very high spatial resolution satellite imagery that was acquired at different points during the growing seasons. This comparison showed that moss biomass was most closely related to early summer NDVI, whereas vascular plant LAI was more closely related to mid-summer NDVI, which suggests that multi-temporal imagery may be useful for quantifying different aspects of plant and soil attributes in tundra ecosystems. The researchers conclude that spatial extrapolation of plant and soil attributes may require the use of land cover maps and field sampling within land cover types rather than linking field measurements directly with remote-sensing observations. In general, the study is robust and multi-faceted, and the manuscript is very well written. Overall, the study makes a valuable contribution to arctic ecology and would be well-suited for Biogeosciences, though could benefit from some refinements detailed below.

General comments (i) I agree with reviewer 1 that the abstract is too long and detailed. The primary findings and implications would be better highlighted if the abstract was condensed. REPLY#16 – We shortened the abstract.

(ii) One of the primary conclusions from this study is that spatial extrapolation of plant and soil attributes will require using land cover maps and field sampling within different land cover types. This approach contrasts with the direct remote sensing approach that involves (1) developing statistical relationship between field and surface reflectance measurements and then (2) modeling plant/soil attributes across a broader area by applying these statistical relationships to wall-to-wall surface reflectance measurements. The authors' conclusion is based on the observation that NDVI often explained little

of the spatial variation in plant and soil attributes across the network of plots. This comparison alone does not seem like an adequate basis from which to draw the conclusion stated above. A direct remote sensing approach does not need to be based solely on NDVI, but rather plant/soil attributes could be predicted using all the spectral bands and plus derived texture metrics and spectral indices. Further analysis might support the authors' current conclusions, but the current conclusion seems premature given the analysis presented. REPLY#17 – We rephrased our conclusions based on the new measures of uncertainty in the LCT approach (P12, L3-8; P15, L23-27). The LCT approach includes all features available in satellite imagery and topography, but still, the difficulty in predicting moss biomass remains (P10, L23-29). Predictions of vascular LAI, shoot mass and soil OM, which is positively related to all plant attributes, instead have relatively low measures of uncertainty.

(iii) The authors estimated the total amount of leaf area, plant biomass, and soil organic mass that occurred within several land cover types found in their study area; however, these numbers do not include estimates of uncertainty. The lack of uncertainty estimates also fits with my comment above (ii). The authors could estimate uncertainty in these totals based on variation in attributes within each land cover type or could perhaps use a Monte Carlo approach in which they account for variation with each land cover type as well as uncertainty in the land cover map. REPLY#18 – We supplemented Table 3 with two types of uncertainty estimates. First, we added standard errors for the estimates of plant and soil attributes in LCTs. Second, we calculated two different estimates – predicted and adjusted – with the help of the LCT map and the classification confusion matrix presented in Supplementary Table 2. To produce predicted estimates, we simply multiplied the percentage cover of a LCT with its field measured mean estimates of vegetation and soil parameters. For adjusted estimates, we took the LCT map uncertainty into account by adjusting the predicted estimate of a LCT with probabilities that the area in concern belongs to other LCTs (e.g. the adjusted estimate of leaf area for shrub tundra is a sum of estimates of leaf areas for all possible LCTs, presented in Supplementary Table 2, weighted by their respective probabilities).

This procedure is now explained in the text (P7, L36 – P8, L3).

(iv) The tea bag decomposition measurements don't seem to fit with any of the stated objectives and are not mentioned in the discussion. The manuscript as already has quite a few elements, so I'd suggest dropping those measurements from the manuscript and focusing on the core elements. REPLY#19 – The tea bag trial is an important part of the study and we now better illustrate the meaning of the results in terms of ecosystem carbon fluxes (P13, L15-28).

(v) It would be good to note in the discussion that the remote sensing observations were not acquired concurrent with field sampling, which introduces uncertainty into the comparisons between NDVI and field measurements. The QuickBird imagery was acquired almost a decade prior to field sampling, whereas the WorldView-2 images were acquired with a year or two of field sampling. These time lags could make it harder to relate field measurements to NDVI, especially for the QuickBird imagery. REPLY#20 – Correct, we added a note of cautiousness in the text (P14, L10-11).

Specific comments (i) P4, L10: Spell out "DD" the first time it is used. REPLY#21 – Done.

(ii) P4, L37-39: Please clarify whether you harvested live vascular shoot biomass, or live + standing dead vascular shoot biomass. Also, please describe how you defined the bottom of the moss layer. REPLY#22 – We clarified these details in the text (P4, L31-34).

(iii) P5, L1: Define leaf area index (m2 leaf m-2 ground) and whether LAI was based on projected leaf area, hemi-leaf area, or two-sided leaf area. It seems you used projected leaf area. REPLY#23 – We defined LAI (P4, L27) and explained that we used projected leaf area (P4, L35).

(iv) P5, L26-27: I'd encourage the authors to put all of the plot-level measurements in the supplemental material, as well as the current summary for each land cover type.

The current summary table should also probably include the standard deviation of each measurement for each land cover type. REPLY#24 – We feel that figures depicting variation among and within LCTs deserve to be included in the main manuscript as capturing and explaining this variation is a key target in our study. In Table 3, we added standard errors to the estimates and also now illustrate the uncertainty in the LCT mapping (see REPLY#18).

(v) P6, L5: You might add that NDVI has been used not only for "spatial examination of LAI" but also for mapping plant aboveground biomass in tundra ecosystem (e.g., Raynolds et al. 2012, Berner et al. 2018). Raynolds: http://www.tandfonline.com/doi/abs/10.1080/01431161.2011.609188 Berner: http://iopscience.iop.org/article/10.1088/1748-9326/aaaa9a . REPLY#25- Good point, we revised the text accordingly (P6, L4-5).

(vi) P6, L8: Technically, you generated a digital surface model (DSM) rather than a digital elevation model (DEM). A DSM includes the height of vegetation and other features, while a DEM represents bare-year elevation. REPLY#26 – We agree that we generated a DSM instead of a digital terrain model (DTM), which does not include vegetation. However, DEM can refer to both DSM and DTM, so we decided to use the term DEM as it is more widely used than the term DSM. Nevertheless, we now mention that our DEM is a DSM instead of a DTM (P6 L9-12).

(vii) P6, L10: Define "ground control point" acronym (GCP) in this sentence, which is the first time the term is mentioned. REPLY#27 – Done.

(viii) P 13, L33-38: What is the range in elevation among the field plots? It is probably quite small. Could it be that topography wasn't a strong predictor of plant/soil attributes because the digital surface model was not accurate enough to differentiate small, but ecologically important differences in elevation among plots? Topography might be a stronger predictor where there is greater topographic variation among field plots. REPLY#28– Elevation ranges from 1 to 20 m. We included this to study area

description (P3, L36). We also now discuss in more detail the potential reasons for the lack of link between moss mass and topography (P15, L4-11).

Tables and figures F1. It is hard for me to differentiate some of the lines used in the figure. Could these be plotted using color, or a variety of line types? REPLY#29 – We produced a new figure with colors.

F3. Spell out "OM" in the figure legend before using the abbreviation. REPLY#30 – Done.

F4. Specify that the plotting symbols represent averages and error bars represent 85% CI. REPLY#31 – This information is given in the legend.

F7. Increase the size of text in the figure. REPLY#32 – Done.

F8. Proved a little more detail in the legend, such as how the land cover map was derived. REPLY#33 – We added more details.

F9. Most of the legend is contained with in parentheses and separated by a bunch of semi-colons and comma. Breaking those five lines into several sentences would make it easier to read. REPLY#34 – We agree and simplified the legend. We also checked other legends.

T2. Perhaps note in the table caption that these numbers are derived using the mul-tiresponse permutation procedure. REPLY#35 – Done.

T3. The column names "early season SOM" and "late season SOM" are somewhat confusing. Maybe it would be clearer to label those two columns as "SOM in unfrozen soil (Gg)" and then have sub-column names labeled "Early season" and "Late season". REPLY#36 – A good suggestion, we revised the table accordingly.

Please also note the supplement to this comment:
https://www.biogeosciences-discuss.net/bg-2017-569/bg-2017-569-AC2-supplement.pdf

---

## Author Response (AR2)

Dear Victor Brovkin,

thank you very much for your positive response to our revised manuscript.

Following your advice, we further reduced the length of the abstract and added conversion factors from plant biomass (separately for each plant functional type) and soil OM to carbon in the Materials and methods section (P5, L22-24), Table 3 legend and Supplementary Table 1 legend.

Yours sincerely,

Juha Mikola